# Reviewing Evidence for the Relationship of EEG Abnormalities and RTT Phenotype Paralleled by Insights from Animal Studies

**DOI:** 10.3390/ijms22105308

**Published:** 2021-05-18

**Authors:** Kirill Smirnov, Tatiana Stroganova, Sophie Molholm, Olga Sysoeva

**Affiliations:** 1Institute of Higher Nervous Activity and Neurophysiology, Russian Academy of Science, 117485 Moscow, Russia; kirillsmirnov@ihna.ru; 2Center for Neurocognitive Research (MEG-Center), Moscow University of Psychology and Education, 123290 Moscow, Russia; stroganova56@mail.ru; 3The Cognitive Neurophysiology Laboratory, Ernest J. Del Monte Institute for Neuroscience, Department of Neuroscience, University of Rochester School of Medicine and Dentistry, Rochester, NY 14642, USA; sophie.molholm@einsteinmed.org; 4The Cognitive Neurophysiology Laboratory, Departments of Pediatrics and Neuroscience, Albert Einstein College of Medicine & Montefiore Medical Center, Bronx, NY 10461, USA

**Keywords:** resting state EEG, spontaneous EEG, Rett syndrome, MECP2, translational biomarker

## Abstract

Rett syndrome (RTT) is a rare neurodevelopmental disorder that is usually caused by mutations of the *MECP2* gene. Patients with RTT suffer from severe deficits in motor, perceptual and cognitive domains. Electroencephalogram (EEG) has provided useful information to clinicians and scientists, from the very first descriptions of RTT, and yet no reliable neurophysiological biomarkers related to the pathophysiology of the disorder or symptom severity have been identified to date. To identify consistently observed and potentially informative EEG characteristics of RTT pathophysiology, and ascertain areas most worthy of further systematic investigation, here we review the literature for EEG abnormalities reported in patients with RTT and in its disease models. While pointing to some promising potential EEG biomarkers of RTT, our review identify areas of need to realize the potential of EEG including (1) quantitative investigation of promising clinical-EEG observations in RTT, e.g., shift of mu rhythm frequency and EEG during sleep; (2) closer alignment of approaches between patients with RTT and its animal models to strengthen the translational significance of the work (e.g., EEG measurements and behavioral states); (3) establishment of large-scale consortium research, to provide adequate Ns to investigate age and genotype effects.

## 1. Introduction

Rett syndrome (RTT) is a neurodevelopmental disorder, occurring in approximately one of 10,000 live female births irrespective of country and ethnicity [1,2,3,4]. It is one of the most common causes of mental retardation in girls. Rett syndrome takes its name after the Austrian psychiatrist, Dr Andreas Rett, who first described girls with this condition in 1966. Since that time, the clinical symptoms of RTT have been refined several times, but still the most prominent feature of RTT is the striking stereotypic hand movements. Other prominent features of the syndrome are cognitive and motor deficits, autistic traits, seizures and dysfunction of autonomic nervous system, including respiratory abnormalities [5,6,7]. The full range of clinical symptoms of RTT and current diagnostic criteria can be found in the review of Neul and colleagues [8].

Rett syndrome is characterized by a developmental progression described as a sequence of age-related stages. Stage I, “early onset stagnation” starts at about 6–18 months of age and can be preceded by a period of seemingly normal development. Stage II, “developmental regression”, occurs at about 1–4 years of age and is characterized by regression of purposeful hand use and spoken language; hand stereotypies and gait abnormalities also occur in this period, during which many girls with RTT are losing their ability to walk, even if they previously acquired this skill. Stage III, the “pseudostationary period”, is the longest stage in RTT. It is characterized by relative stabilization of clinical manifestations and even slight improvements in cognitive and communicative domains, accompanied by slow regression in neuromotor functions. Stage IV, “late motor deterioration”, is also reported in girls with RTT where pronounced motor deterioration eventually leads to death. According to recent views this stage can be shifted to late adulthood or even fully avoided though rehabilitative therapy [9,10].

In 1999 mutations in the X-linked gene *MECP2*, which encodes the transcriptional regulator methyl-CpG-binding protein 2 (MECP2), was found to be the main cause of RTT [11]. The most common de novo *MECP2* mutations in patients with RTT are R106W, R133C, T158M, R168X, R255X, R270X, R294X and R306C, which are responsible in total for about 60% of RTT cases and C-terminal deletions (9%) and large deletions (8.5%) [12,13,14]. Discovery of the role of the *Mecp2* gene in RTT and the fact that it is highly conserved in vertebrates allowed for the development of animal models, providing critical insight into the pathological processes, and testing of potential treatments. The initial models were male mice with a total knockout of the *Mecp2* gene and hence complete disruption of the production of the MECP2 protein [15,16]. Although RTT in humans is caused by less extensive disturbances in MECP2 structure, these models recapitulate many of the features of Rett syndrome including pronounced stereotypic forelimb motions, uncoordinated gait, reduced spontaneous movement and irregular breathing.

Methodological advances have allowed the development of several female animal models with local mutations recapitulating the specific variants of human RTT such as R168X [17], T158A [18], R255X [19] and A140V [20]. Still, these models are relatively recent, and the full knockout male mouse model dominated in the literature to date, possibly due to earlier symptom onset and a more pronounced phenotype. Additional models of MECP2 abnormalities include the heterologous expression of human *MECP2* in *Drosophila* [21]; the disruption of *Mecp2* expression in zebrafish [22,23]; *Mecp2* knockout in rats [24,25]. Cellular cultures that use stem cells from mouse models of RTT [26] or human induced pluripotent stem cells (hIPSC) [27] are another perspective way to investigate how *MECP2* disruption affects neuronal development, including impaired proliferation and proneness to senescence [28].

An important feature of face validity of animal models of RTT is the presence of a period of relatively normal development followed by regression. Studies in presymptomatic animals and animals in the beginning of regression can shed light on the primary mechanisms that are disrupted first and might trigger the subsequent abnormalities. To assess disease severity in animal models of RTT, the scale is used in which inertia, gait, hind-limb clasping, tremor, irregular breathing and poor general condition are scored from 0 to 2 (0—the symptom is absent, 1—symptom is present and 2—the symptom is severe) (Guy et al., 2007). Male mice with T158A mutation in *Mecp2* gene start to develop symptoms after 4 weeks of life, while female mice with the same mutation develop symptoms after the 16th week [29]. In relation to human lifespan, adolescence in mice lasts from postnatal week 3 to 9 (with sexual maturation starting around day 47); adulthood ends approximately around week 107 and is followed by the post reproductive stage [30]. Thus, males of the animal model of RTT develop a disease phenotype during adolescence before puberty and females become symptomatic at the young adult stage. 

Animal research has revealed the multifaceted role of MECP2 in cell functioning. Initially defined as a transcriptional repressor, MECP2 was then found not only to repress but also to activate transcription, to modify chromatin structure and affect miRNA processing. MECP2 regulates gene expression in a subtle but widespread manner affecting thousands of genes [31]. Transcriptomic methods showed that the influence of *MECP2* mutations is extensive, affecting over 60 molecular pathways, including PI3/Akt/mTOR signaling pathway, important for dendritic complexity, soma size and spine morphology [32]. Noteworthy, two other genes, also identified to cause RTT, cyclin-dependent kinase-like 5 (*CDKL5*) and forkhead box protein G1 (*FOXG1*), are thought to converge on MECP2 [33,34]: MECP2 is an enzymatic target of CDKL5, while FOXG1 interacts with the MECP2-isoform (MECP2-e2; one of two MECP2-isoforms) that prevents cell death of cerebellar neurons [35]. FOXG1 and MECP2 also have overlapping expressions in differentiating forebrain cortex [33]. Animal models have been used to gain understanding of neural dysfunction and the genesis of seizures in RTT. We will review these EEG data under the subhead Animal Studies in the results section.

Electrical brain activity in the form of EEG can be recorded both in humans and animals and might serve as common neural biomarkers that reflect specific neural network dysfunction and provide validity of between-species translation. Electroencephalography (EEG) reflects summed slow dendritic potentials of many cortical pyramidal neurons. EEG rhythms in different frequency bands emerge from the dynamic interaction between neuronal populations and have been associated with a range of different cognitive processes [36,37]. Thus, recordings of spontaneous EEG (i.e., EEG recorded in the absence of controlled stimulation or task conditions and not stimulus or task locked) can be used to make inferences about abnormalities in brain functioning. The first description of Rett syndrome by Dr. Rett in 1966 contains reference to the atypical EEG patterns that were later detailed in the seminal work of Hagberg [38]. Distinct EEG patterns were described for the different stages of RTT, with a “characteristic EEG development” suggested as one of the supportive criteria for RTT diagnosis [5,39]. However, well-characterized EEG activity with known neurophysiological underpinnings are still lacking for RTT. Here we provide a review of studies on spontaneous EEG with the goal of identifying potential EEG biomarkers of MECP2 dysfunction. We draw on animal studies to capitalize on the in depth molecular and neurophysiological data that have been collected using invasive approaches and connect this with available non-invasive EEG data from patients with RTT. We assess correspondence between animal and human EEG, and how these can be used to understand the underlying pathological processes assessed by more invasive measures in animal models. We believe this is a critical step to advancing the potential of preclinical studies.

For this review we focus on papers reporting data collected during recordings of spontaneous EEG, which constitutes the majority of studies that applied the EEG method to patients with RTT since they are commonly made in the clinic to assess the presence of clinically relevant EEG abnormalities. Evidence of atypicalities in the event-related potentials in RTT are summarized elsewhere [40]. 

## 2. Results

The results are organized into the following sections: 1. epilepsy/seizures; 2. abnormalities in resting EEG spectra and their neurophysiological underpinnings; 3. EEG abnormalities associated with sleep disturbances, and their neurophysiological underpinnings; 4. seizures and EEG abnormalities through developmental changes; 5. behavioral and EEG abnormalities in relation to RTT genotype; 6. EEG as a biomarker of treatment efficacy. These are subdivided into human studies and animal studies. Finally, we summarize and integrate these findings in a conclusion section.

### 2.1. Epilepsy/Seizures

#### 2.1.1. Patient Studies

##### Seizure Semiology

Most studies resulting from our search concerned epileptiform abnormalities/seizures in RTT (Figure 1, Table 1). Seizures of various semiology were described, with the most common being generalized seizures, tonic-clonic seizures and complex partial seizures [41,42,43,44,45]. Reports on the prevalence of epilepsy in patients with RTT ranged between 50 and 90%, and about 30–50% of cases were drug-resistant [41,44,46,47,48,49,50,51,52,53]. This wide range of results might be related to different methods used to identify seizures. Notably, many authors agreed on the fact that in patients with RTT, seizures are difficult to distinguish clinically from disorganized movements such as hand stereotypies, tremor and dystonia, or breathing abnormalities, which are typical in RTT [41,43,46,51,54,55,56,57,58]. While not always feasible, future work in which long-term video EEG polygraphy is used for seizure diagnostics would yield more reliable results. At the same time, there are a significant number of RTT cases with seizure-like abnormalities manifested behaviorally, but not in EEG (so-called vacant spells). Whether it is a detectability issue, or these vacant spells are indeed unrelated to epileptiform activity and represent peculiar behavioral phenomena remains to be examined in future studies.

Unlike plenty of large-sample clinical studies on epilepsy/seizures (generally *n* > 100), quite a few studies [38,48,49,55,62,63,73,80,81,83,87] provided description of epileptiform activity in RTT patients with reference to EEG recording (with largest *n* = 78). These studies reported a much higher incidence of epileptiform activity than epilepsy in patients with RTT. In the two largest studies, epileptiform activity was present in 78–82% of patients with RTT [74,84], while epilepsy was confirmed only in 50–76% of these patients. More recent study of Buoni and colleagues reported even higher incidence of epileptiform activity in RTT as each of their 16 RTT patients has some form of epileptiform abnormalities, while clinical seizures were observed in only 6 patients [50]. A broad range of baseline epileptiform activity was reported in patients with RTT, such as single or grouped spikes and sharp wave discharges, often followed by relative attenuation of background activity, slow spike and wave, or continuous spike and wave [44,55,63,80,87,90]. The most frequently reported epileptiform activity in RTT is centro-temporal spikes (CTS), which we will return to in a subsequent section.

##### Centro-Temporal Spikes as a Frequent Epileptiform Abnormality in RTT

Among the broad range of epileptiform abnormalities seen in RTT, centro-temporal spikes (CTS) of short duration are the most common and have the earliest manifestation [38,80,90]. CTS are particularly associated with idiopathic focal epilepsy syndrome in childhood (aka, benign epilepsy of childhood with CTS: BECTS). BECTS is the most common pediatric epilepsy syndrome, usually occurring in asymptomatic children aged 4–15 years, with cessation of seizures by adulthood. In contrast to the previously assumed benign course, a growing literature has documented pervasive cognitive and/or behavioral problems in children with BECTS, especially in the domains of language, verbal memory and fine motor skills [91,92].

CTS can be suggested to reflect altered excitability of the perisylvian cortical areas that may contribute to motor and speech disturbances in RTT patients. Evidence for the relevance of CTS to motor control comes from studies that found a suppression of focal CTS by hand tapping/clapping in a subgroup of patients with RTT with six representative cases described in the literature [60,61,75,93]. In line with these observations, Cooper and colleagues reported a decrease in EEG abnormality during hyperventilation that is accompanied by increased movement in young girls with RTT [74]. The suppression of CTS by hand movements is a rare phenomenon, and we are aware of only one other neurological condition, in which this phenomenon was also described-BECTS [93]. The authors suggested that decreased top-down inhibitory control over the motor cortex is the main cause for such motor cortex hyperexcitability at rest [80,94]. In line with this hypothesis, frontal hypoperfusion was reported in patients with RTT [75] and recent research demonstrated that BECTS is associated with decreased fMRI functional connectivity in dorsal frontal network [95], and with a deficit in executive functioning [96]. Niedermeyer’s suggestion is also in accord with recent animal findings of reduced excitatory synaptic connectivity in the prefrontal cortex of mice lacking MECP2 [97], and with evidence proving a causal role of disrupted *inhibitory transmission* in RTT animal models [98].

Interestingly, some patients with RTT exhibit the opposite relationship between CTS and movements, such that centro-temporal spikes in their EEG are triggered by hand movements and disappear when movements stop. Hand stereotypes were a major trigger of this epileptiform activity with six patients introduced in case reports [59,65]. Hand stereotypes often coincides with spikes over central region with passive tactile stimulation leading to its facilitation in about 30% of patients with RTT as reported in the first EEG studies on RTT (2 out of 7 patients in [85] and in 9 of 26 in [84]), suggesting high incidence of this response in RTT. Noteworthy, in some cases CTS can be precipitated by specific movements. In a case reported by Luo and colleagues, CTS were only induced by right hand-to-lips tapping but not by right hand tapping cheek or abdomen, the left hand tapping lips or observer’s hand tapping the lips [60]. Roche Martínez and colleagues reported three epileptic Rett patients with reflex seizures triggered by food intake or proprioception [66]. Conceivably, unlike those CTSs that are suppressed by movements, CTSs that are triggered by movements may be related to enhanced excitability of somato-sensory cortex, which receives proprioceptive feedback during movement execution. Indeed, some patients with RTT display exaggerated cortical somato-sensory responses to incoming somatosensory input in the form of “giant somatosensory potentials” [99,100,101]. The “giant somatosensory potentials” in RTT patients are similar to those in patients with cortical reflex myoclonus, which are often associated with myoclonic jerks [55,67,99]. Reduced phasic inhibition, mediated by GABAa receptor activity, that is described in the subsequent session on animal models, might also play a role in the generation of this seizure type.

Thus, centro-temporal spikes in RTT seem to be related to movement atypicalities, although how they are induced or ameliorated across participants varies, suggesting different paths to this movement associated epileptic activity. Excitatory/inhibitory balance seems to be altered in different ways in various parts of the brain and even cortical areas. Type of mutation and X-inactivation profile may play major roles in the relationship between reduced top-down control from prefrontal cortex and hyperexcitability of somatosensory cortex in RTT patents determining the mechanism of CTS induction and suppression.

##### Development of Seizures and Severity of RTT Symptoms

Average age of seizure onset in patients with RTT is about 4 years of age, that is greater as compared to other patients with severe mental retardation, who develop seizure much earlier before children reach their first birthday [102]. At the same time, earlier seizure onset in RTT was associated with a more severe course of seizure but not with the severity of the other RTT symptoms [103]. Many patients with RTT are seizure-free at stage II when the clear symptoms of the deterioration in motor and cognitive behavior are already present.

Nonetheless, there is some evidence for association of epileptic seizures with greater severity of RTT. Pintaudi and colleagues showed that less severe forms of the condition, characterized by the preservation of speech, is associated with lower incidence of epilepsy than classic forms of the syndrome [44]. Tarquinio and colleagues reported that another atypical form of RTT with more profound developmental delay exhibited higher prevalence of seizures [43]. In line with this link between RTT severity and epilepsy, patients with a classic RTT phenotype who are more severely affected are more likely to have seizures [43,51,104]. As standardized measures of cognition cannot typically be used to assess cognitive function in individuals with RTT, we highlight the study of Vignolli and colleagues [49] that implicated eye-tracking to objectively evaluate cognitive performance of 18 girls with RTT at clinical stage III and IV on several cognitive tasks. For example, voluntary gaze behavior served to demonstrate the ability to follow verbal instructions, and recognition and categorization of visual images. Performance on these tasks, and more traditional measures of behavioral functioning (vineland adaptive behavior scales) were directly compared with severity of epilepsy and EEG epileptiform abnormalities in the RTT sample. The authors reported that earlier age of epilepsy onset and higher seizure frequency were associated with poorer cognitive outcome (fixation time on correct stimuli). In addition to these clinical features, relatively worse cognitive functioning in girls with RTT was also associated with severely disturbed spontaneous EEG patterns that were characterized by more diffuse and multifocal epileptiform discharges. In accord with this conclusion, Buoni and colleagues [50] observed that multifocal epileptiform discharges were absent in EEG of high functioning girls with RTT with preserved speech, although they still had focal EEG centro-temporal spikes. Nevertheless, causal relationships between seizures and developmental trajectory of RTT disorder remains unclear. For example, a recent large-scale study of 793 individuals reported that there were no significant associations between age of developmental regression and age of seizure onset in patients with RTT or in patients with so-called RTT-like disorders [42]. Thus, findings of the above correlative nature do not ascertain whether the epilepsy plays a crucial role in deteriorated cognitive abilities in RTT, or if it alternatively reflects the more severe underlying pathophysiology [105]. In support of the latter explanation, Cooper and colleagues reported that increases in EEG abnormalities and seizures follows developmental regression rather than precedes it in patients with RTT, thus reflecting the consequences of pathological processes that trigger the regression [74]. In sum, epilepsy is hardly an underlying cause, rather it displays comorbidity with RTT disorder. While at the extremes (e.g., when comparing atypical forms of RTT) it seems that seizure frequency relates to RTT severity and this is certainly of interest, by itself it is unlikely to be a valuable biomarker of RTT. 

#### 2.1.2. Animal Studies

Animal models of RTT partly recapitulate a seizure phenotype described in patients with RTT [106,107,108]. The most confirmed form of seizures in mice with MECP2 deficiency are spontaneous absences accompanied by spike-wave discharges (SWDs) [107,109,110,111,112], which undergo dramatic increase in incidence and duration from 13 to 104 weeks [113] (Table 2). SWDs observed in mouse models of RTT have a frequency of around 6–8 Hz and are suppressed by ethosuximide, which is similar to findings in inbred strains (i.e., nearly genetically identical animals due to long inbreeding), which were used to create RTT models [114]. At the same time in comparison to inbred strains, RTT mouse models have increased number of SWD episodes and prolonged duration of these episodes pointing to the modulatory role of MECP2 deficit in the development of this type of seizures. Despite being the most prevalent type of epileptic seizures in animal models of RTT, typical absences are rarely observed in human patients with RTT [41]. Partly it could be explained by difficulties in their detection in patients as this type of seizures can be easily missed without prolonged EEG recording. Another explanation might be related to between-species differences in thalamocortical interactions and the role of sensory inflow: in rodents, sensory inflow from whiskers into somatosensory cortex, the dominant area of SWD, is crucial for generation of absences [115]. Differences between epileptic phenomenology in human patients and animal models also can be explained by the much more pronounced Mecp2 abnormalities in the most common animal models of RTT–*Mecp2*-knockout male mice as compared to female patients with *MECP2* mutations. In line with this explanation, study on male mice with MECP2 truncated at C-terminal domain found myoclonic jerks accompanied by single SWDs [106]. Thus, the milder Mecp2-dysfunction may lead to the generation of single SWDs but in contrast to *Mecp2*-null mice prevents the development of full-blown absences with SWD complexes. Noteworthy, myoclonus is frequently reported in patients with RTT [55,67,99].

Nevertheless, there is a question why and how MECP2 deficiency leads to the increase in the occurrence of seizures. Lang and colleagues reinstated MECP2 expression in *Mecp2*-deficient adult mice and observed a significant attenuation of abundant SWD in their spontaneous EEG [111]. Thus, even in adult subjects constitutively lacking *Mecp2* the absence seizures may be abolished through restoring a normal MECP2 function in the whole brain. More specifically, the restoration of MECP2 function specifically in GABAergic neurons rescues the seizure phenotype [120], suggesting that abnormal inhibitory transmission is both necessary and sufficient for absence seizures induction in RTT. The selective removal of *Mecp2* from glutamatergic neurons also induces absence seizures but only in 37.5% of animals, and the restoration of MECP2 expression only in these neurons does not lead to freedom from seizures [116]. Thus, abnormal inhibitory transmission has a primary role in absence seizure generation, while impaired excitation may also contribute to this process but probably in an indirect way through activation of inhibitory neurons [98]. Although not being necessary for development of absences, altered excitation may be causative for convulsive seizures. Activation of kainate receptors leads to increased power of gamma oscillations in *Mecp2*-null mice in comparison to wild type in vitro and induces more pronounced behavioral seizures in vivo [121].

Closer look into GABAergic impairment in MECP2-deficient animals can explain the occurrence and specificity of epileptic activity (Figure 1). MECP2 is highly expressed in GABAergic neurons (150% compared to neurons lacking GABA) and the loss-of-function of MECP2 leads to decreased intracellular concentration of GABA [119]. Proteins involved in GABA synthesis (GAD65, GAD67), vesicular packing (VGAT) and reuptake (GAT1) are downregulated in animal models of RTT [108,119,122]. Increased GABA spillover due to downregulated GAT1 induces activation of extrasynaptic GABA receptors, which increases tonic inhibition. At the same time, the increased tonic inhibition was shown to be associated with the development of SWDs [123,124]. In line with this logic, Zhang and colleagues proved that in animal model of RTT increased spillover concentration of GABA is consequential for seizures, as application of GAT-1 inhibitor further increases epileptiform discharges in MECP2-deficient mice, and at a higher dose induces the epileptiform discharges even in the wild type animals [108]. In firm support of a causative role of increased tonic inhibition for seizure generation in mice lacking MECP2, this study demonstrated that enhancing a sensitivity of GABAb receptors by acting on them with selective activity-dependent receptor enhancer increased seizure incidence and duration but slowed down the frequency of oscillations, while agonist of extrasynaptic GABAa receptors decreased their incidence. Such changes are well-predicted by a computational model in which increased GABAb receptors’ conductance led to a higher incidence, a longer duration and lower frequency of SWDs and the opposite effects were observed after increasing activation of GABAa receptors [125]. The extend of *Mecp2* dysfunction was also found to be related to SWD characteristics: male mice RTT model, with full absence of X-linked *Mecp2*, have longer duration and slower frequency of SWD than females which have the intact Mecp2 gene on one X chromosome [111]. Thus, characteristics of SWD may serve as a biomarker of GABAergic impairment in animal models of RTT. As for RTT patients, more attention to detection of SWD in this group will clarify the true prevalence and etiology of this epileptiform activity in RTT patients. In line with causative role of imbalance between extrasynaptic GABAa and GABAb receptors in the development of SWDs, Okamoto and colleagues described the female patient with mutation in gene encoding extrasynaptic GABAa δ-subunit, which was diagnosed with RTT and showed SWD [126].

Tactile stimulation by routine handling evokes convulsive seizure in animals lacking MECP2 in GABAergic neurons [109]. These seizures were observed at age of 12 weeks when behavioral symptoms were already pronounced resembling the development of epilepsy in RTT patients, which is usually diagnosed after the developmental regression [46]. Ito-Ishida and colleagues showed that handling induced tonic-clonic seizures only after conditional deletion of *Mecp2* from somatostatin-positive but not parvalbumin-positive inhibitory neurons [110]. Meng and colleagues observed similar tonic-clonic seizures in mice with selective preservation of MECP2 in glutamatergic neurons, which underlines the crucial role of MECP2 in inhibitory neurons in the development of this type of seizures in animal models of RTT [116]. Induction of seizures by handling can be caused by evoked hyperexcitability of neuronal networks due to reduced phasic inhibition. Indeed, application of GABAa antagonist bicuculline leads to higher incidence and duration of epileptiform spikes in hippocampal slices [121]. Noteworthy, GABAa mediated tonic inhibition was also decreased in the hippocampus, which can also participate in proneness to convulsive seizures [127]. 

The screwed crosstalk between GABAa and GABAb in RTT might be related to potassium chloride cotransporter 2 (KCC2), which contributes to the low intracellular chloride concentrations found in mature neurons and thus establishes the conditions for the hyperpolarizing effect of GABAaRs [128]. KCC2 is downregulated in RTT patients and mouse models [129,130,131]. Activation of the GABAbR results in reduced levels of KCC2 at the cell surface, which parallels an increase in intracellular chloride, thus making GABAa-mediated inhibition less efficient [132]. Affecting the ratio of KCC2 to Na-K-Cl Cotransporter 1 (NKCC1), a membrane transporter that is functionally antagonized to KCC2, seems to be a promising direction in RTT therapy restoring electrophysiological and morphological characteristics in MECP2-deficient animals and neurons cultured from RTT patients [133,134].

Besides the inhibitory transmission by itself, a preponderance of seizures in RTT is affected by a plethora of factors modulating the inner excitatory state of inhibitory circuitry. For example, seizure predisposition may be provoked by a selective loss of MECP2 in cholinergic neurons via decreased expression of excitatory cholinergic receptors (α7 nAChR) in parvalbumin-positive inhibitory neurons [135]. Thus, the genesis of seizures in RTT can be attributed to a multilevel disruption of neuronal circuits consequential to the dysfunction in several different molecular pathways with the major role of inhibitory signaling, particularly the balance between tonic and phasic inhibition.

### 2.2. Abnormalities in Resting EEG Spectra and Their Neurophysiological Underpinnings

#### 2.2.1. Patient Studies

##### General Slowing of Background EEG Activity

EEG abnormalities in RTT are not limited to epileptiform activity. General slowing of background EEG activity was consistently reported in patients with RTT and is observed even at the earliest stage of the disorder [39,49,76,79,81,84,87,94,136,137]. General slowing of EEG rhythms in clinical studies is often referring to as one of two qualitatively distinct phenomena. This term designates either an abnormally low peak frequency of oscillations within particular functionally distinct EEG rhythm e.g., sensory-motor mu rhythm or visual alpha rhythm, or heightened amplitude and/or abundance of theta and delta waves in spontaneous EEG in combination with abnormally low prominence of higher frequency activity within alpha and beta frequency bands. Both phenomena have been described for EEG of patients with RTT [49,71,76,80,81,94,99,137].

Several authors reported that the sensory-motor mu rhythm, with typical functional reactivity to passive/active movements and topographical maximum at central scalp electrodes, had abnormally low frequency in patients with RTT [76,137]. The shift of mu rhythm towards lower frequency was on average almost 5 Hz, with mu frequencies varying from 8 to 13 Hz in typical subjects to 4–6 Hz in some patients with RTT. However, since none of the above studies employed statistical between-group comparisons, it is unclear whether “mu rhythm slowing” represents a general feature of RTT disorder or if it rather characterizes a specific subset of patients. Regarding visual alpha rhythm, occasional observations suggest that alpha frequency in people with RTT varies from unremarkable posterior alpha rhythm [76] to abnormally slow alpha variant [81]. However, the specific characteristics of these phenomena have not been quantified to date. Clearly, rigorous investigations into the spectral characteristics, cortical topography and functional reactivity of alpha rhythms in somatosensory and visual modalities are needed to confirm and extend the available qualitative observations of their abnormalities in people with RTT. Linking these EEG features to distinct RTT phenotypes appears to be a promising area of future research.

A more frequently reported EEG abnormality in RTT is atypical prevalence of low frequency activity in EEG spectra. This abnormality was well characterized both by clinical description [71,82,87,89,90,136] and by statistical analysis (Table 1 and Table 3). Since the first clinical reports EEG slowing was consistently reported in girls with RTT, e.g., in the form of delta and theta bursts [89,136]. Abnormal theta activity may be present either in the form of highly regular and prolonged fronto-temporal rhythmic episodes, or as the diffuse fronto-temporal theta waves with amplitudes twice as much as the background activity in the resting EEG [49,71,76,80,81,87,99,137,138]. 

Quantitative studies of spontaneous EEG recorded under eyes-open or eyes-closed conditions show the attenuation of alpha and beta band power with concomitant generalized increase of theta and delta activity in RTT [137,139,142,143]. Moreover, abnormally increased delta activity in girls with RTT was reported also during slow wave sleep (SWS), suggesting that predominance of low-frequency activity might characterize EEG in RTT regardless of the functional state [141].

RTT patients with highly deteriorated speech and motor functions were reported to have greater theta relative power across a variety of scalp electrodes [71]. Gratchev and colleagues found that earlier onset of RTT was associated not only with worse behavioral outcome but also with greater reduction of central and occipital alpha power and greater increase in central and frontal theta activity [138]. The largest quantitative study to date [139] found that increased delta power in 57 patients with RTT was correlated with worse performance on the Mullen scales of early learning (MSEL; [144]), a measure of multiple aspects of cognitive functioning). Vignolli and colleagues reported that RTT girls with relatively worse cognitive functioning had more severely disturbed spontaneous EEG patterns with an absence of occipitally dominant rhythmic activity (while not specified likely referring to alpha rhythms), and a marked slowing of background activity [49]. These studies provide tentative evidence for the extent of atypical low frequency activity in “background” EEG tracking with the severity of the disorder. However, since low frequency activity in EEG is not specific to RTT and characterizes a number of other neurological conditions and pervasive developmental disorders (e.g., attention deficit hyperactivity disorder: [145], obsessive compulsive disorder: [146]; Parkinson’s disease: [147]), this potential biomarker of RTT severity might be related to general deterioration of brain function.

##### Spectral Slope

EEG power spectral density is characterized by a negative slope if presented in the semi-log or log–log space, because power of the signal frequency content decreases rapidly as a function of the frequency. Steepness of the 1/f spectral slope correlates with slowing of the background EEG, reflecting prevalence of low frequency EEG over high frequency signals. According to recently developed dynamic neuronal communication framework, more negative 1/f slope of EEG power spectral density represents more synchronous activity of the neuronal population, while flattened (less negative) spectral power slope is related to higher background rates of neuronal firing decoupled from an oscillatory carrier frequency-noise [148]. Higher neuronal noise indexed by a flatter 1/f slope was associated with aging, while pathological overcoupling of neuronal oscillations reflected in the steeper 1/f slope was seen in several neurological and developmental disorders, such as Parkinson’s disease and attention deficits and hyperactivity disorder (ADHD) [148]. Notably, in a study of ADHD, 1/f slope correlated with theta to beta ratio but had greater power to differentiate ADHD from controls. The authors suggested that the novel measure of 1/f slope was a better reflection of underlying pathophysiology over the traditional indexes of EEG background slowing [149]. Given a proposed dependency of a steeper 1/f slope on a decrease in the local excitation/inhibition ratio [148], this EEG measure is of value to researchers of RTT. However, 1/f slope of EEG spectral density was assessed in only one, but large-scale study conducted on 57 RTT patients [139]. Roche and colleagues showed that girls with RTT had significantly more negative (steeper) 1/f slope than their typically-developing peers and the steepness of the slope was correlated with decreases in cognitive functioning as measured with MSEL. Taking into account this promising finding, 1/f slope might be suggested as a potential biomarker of altered neural excitation/inhibition balance in RTT. 

##### Cortical Gamma Oscillations (30–100 Hz)

Cortical gamma oscillations (30–100 Hz) are generated in recurrent circuits of excitatory and inhibitory neurons [150,151] and their amplitude reflects the excitatory state of the neural network. In humans and animals these oscillations are most reliably induced in response to sensory stimuli [152], but might be also recorded in the spontaneous mode, both during wakefulness and sleep [153]. The animal studies revealed that gamma oscillations contain at least two functionally different sub-bands, with a conventional band of low- and high-frequency gamma bands being 30–60 Hz and >70 Hz respectively [154,155,156]. In humans, gamma band activity is usually measured between 30 and 50 Hz in scalp EEG recordings without division into functionally distinct gamma sub-bands. The human EEG gamma band can be contaminated by myogenic artifacts having the same frequencies as neural gamma activity [157]. Despite these challenges, spontaneous gamma oscillations were consistently found to be abnormally enhanced in awake patients with increased E/I ratio, e.g., patients with ASD, schizophrenia and epilepsy [158,159,160]. Studies of spontaneous gamma-band activity in patients with RTT have been limited to relatively low gamma frequency [139,141]. Ammanuel and colleagues found that in patients with RTT (*n* = 10) the power of 35–45 Hz oscillations during slow wave sleep (SWS) did not follow a typical and highly significant decrease from 2–5 to 6–9 years of age, suggesting that subjects with RTT do not exhibit a normal developmental reduction in cortical excitability [141]. In contrast to sleep state, gamma band activity of 30–50 Hz recorded during wakefulness was unremarkable in a sample of 57 girls with RTT of similar age when they watched self-chosen video [139]. Taking into account the decreased muscle tone during sleep as compared to awake state, sleep EEG recordings might be much better suited than awake EEG for detecting “true” gamma activity, especially in patients with muscle dystonia and stereotypic movements since this can contaminate the signal. While the low gamma-band abnormalities during SWS might have the potential to serve as objective biomarkers of increased cortical excitability in RTT, they need to be independently replicated in a larger sample. In addition, specificity to this functional state needs to be further investigated in the future studies.

##### EEG Abnormalities in Relation to the Developmental Stage

As mentioned in the introduction, patients with RTT are characterized by specific patterns of developmental EEG changes that depend on the progression of the disease. This knowledge can be used as an aid in the RTT diagnosis and can help to stage the disorder [5,39]. At stage I of the disorder no clear EEG abnormalities have been reported, probably because this period is largely understudied due to a lack of clear symptoms at this initial stage. However, some rare evidence suggests that epileptogenic activity might already be present at this early stage of the disorder [74]. At the same time, EEG abnormalities might be the only pathological laboratory investigation in patients with RTT at the time when the diagnosis is made (see e.g., [90]). EEG abnormalities usually described during stage II of RTT disorder contain focal spikes with centro-temporal scalp topography (CTS), and/or focal sharp-wave discharges [48,81]. Stage II is also characterized by marked slowing of resting EEG oscillations that progresses further in stage III. Multifocal spikes and sharp-wave discharges also appear at stage III. No occipital dominant activity might be present in awake EEG [83,86]. Intriguingly, these EEG abnormalities might become either more pronounced or, to the contrary, ameliorate at stage IV—the most advanced stage of the disorder [48,68,71,81,83,161,162]. Despite the above described general tendency in EEG development in RTT, there is considerable interindividual variation of EEG patterns, especially in clinical stages III and IV [77]. At these advanced stages spontaneous EEG vary from a “normal” pattern without epileptiform discharges to some rare, very abnormal variants [48,68,81]. For example, Izhizaki and colleagues described a RTT girl with awake EEG at age 23 dominated by monotonous theta rhythm activity that was not influenced by either opening or closing of the eyes but attenuated only by a loud noise or strong pain stimulus [83]. Future large-scale experimental longitudinal studies are needed to identify different trajectories of EEG development and the potential for prognostic value of spontaneous EEG patterns at the earliest stages of the RTT disorder.

The only study that quantitatively examined EEG in girls with RTT during the active regression stage (stage II, *n* = 20) revealed interesting results [139]. While the higher-frequency activity ranging from 6 to 30 Hz was attenuated similarly in patients with RTT regardless of disease stage, the low frequency activity differentiated between the RTT stages. In contrast to widely reported increased power of low frequency activity in patients with RTT [137,142,143], girls with RTT in the active regression period were characterized by lower frontal and central theta and delta activity as compared to typically-developing peers. Decrease in frontal theta activity was so dramatic that about half of the patients had theta power values below the lowest value in the control group, clearly indicative of underlying neuropathology (Figure 3A in [139]). The developmental course of low-frequency activity was also different in patients with RTT and TD: unlike the typical decrease that is seen in the power of delta and theta activity with age, girls with RTT show an increase in delta and theta power with age. Thus, studies suggest atypical developmental trajectory of low-frequency EEG activity with initially low delta and theta activity in girls with RTT for the active regression period and subsequent increase in this activity in the post-regression period.

The results are summarized in Figure 2. The primary findings from human studies are the general slowing of background EEG, that might be also characterized with a new measure of 1/f spectral slope, related to decreased E/I imbalance at rest corresponding to increased tonic inhibition suggested from animal studies. At the same time, not only predominance of low over high frequency oscillation might contribute to this background EEG slowing, but also a shift towards lower values of dominant frequency within particular functionally distinct EEG rhythms, e.g., sensory-motor mu rhythm or visual alpha rhythm. More studies are needed to quantify these shifts, especially related to motor mu-rhythm as motor problems are among the core deficits in RTT.

#### 2.2.2. Animal Studies

Animal EEG studies in waking resting state have been limited to gamma band activity (Table 2). Consistent with observations on interrelations between excessive high frequency gamma (>70 Hz) and epileptogenesis [160,163,164], symptomatic stage male mice with T158A mutation and total deficiency of MECP2 were characterized by atypically increased power of spontaneous high frequency gamma oscillations in waking EEG suggestive of constitutive cortical hyperexcitability [29]. Abnormally exacerbated high frequency oscillations were absent at a presymptomatic stage, which resembles findings from human patients who usually develop epileptic phenotype after regression. Unlike high frequency gamma, power of low frequency gamma in the awake state of the same mice was indistinguishable from that in wild type mice, for both symptomatic and presymptomatic stages.

Spontaneous EEG rhythms in MECP2-deficient mice were also studied during exploratory behavior. They found that frequency of hippocampal theta activity in RTT was shifted toward abnormally low values [107,111,118]. Given that theta frequency is strongly modulated by drugs stimulating catecholaminergic and especially dopaminergic systems [165,166], that are known to play a major role in exploratory locomotor behavior, theta deceleration in RTT may be linked to the deficiency of neuromodulatory influences on the hippocampal activity. In line with this idea, preservation of MECP2 expression only in catecholaminergic neurons in MECP2-deficient male mice resulted in normalization of theta frequency almost up to the wild-type level, in parallel with improvements in ambulatory rate, motor coordination and nest-building [118]. Remarkably, abnormally attenuated exploratory behavior in mice lacking MECP2 and decelerated theta oscillations has been accompanied by atypically decreased power of hippocampal low frequency gamma [111,118]. Coupled theta-gamma dynamics is explained by the fact that in all mammals including humans, low frequency gamma oscillations are modulated by concurrent theta rhythm, whose frequency and amplitude increase during exploratory behavior [154,155,167]. This means that a deceleration of cortico-hippocampal theta and decreased low frequency gamma power in RTT individuals might index deficient catecholaminergic modulation that results in decline of their behavioral state and attenuates locomotor activity. 

To sum up, there are very few studies on spectral analysis of EEG in animal models of RTT and findings are difficult to reconcile. The slowing of spontaneous EEG rhythms has been reported in both animal and human studies; however, different behavioral states were studied: exploration in animals vs. resting state in humans. The altered parameters of spontaneous gamma oscillations in RTT animals makes them an attractive target to pursue in human research, however a few methodological challenges dampen their translational potential. EEG studies during sleep and MEG might overcome some of these challenges. Future translational research is needed to clarify the neurophysiological basis of these clinically relevant features of EEG.

### 2.3. EEG Abnormalities Associated with Sleep Disturbances, and Their Neurophysiological Underpinnings

#### 2.3.1. Patient Studies

Spontaneous EEG activity is influenced by many factors, such as a functional state, level of alertness, mobility and mood. These parameters are difficult to control for, especially in children with neurodevelopmental disorders. Therefore, any comparisons of spontaneous EEG rhythms between RTT and control groups should be treated with caution. EEG recordings during sleep are more immune to behavioral variability and, therefore, provide a compelling option for investigation into altered dynamics of spontaneous EEG oscillations in RTT. When characterizing epileptiform activity, sleep EEG also presents an advantage as epileptiform discharges in RTT patients prone to seizures are identified with higher probability in sleep EEG than in awake EEG [38,68,79,84,88,136,162]. 

Sleep-associated problems are commonly reported in patients with RTT and serve as supportive evidence for a RTT diagnosis [8]. They include dysregulation of the sleep/wake cycle with irregular sleep onset time, abnormally long daytime sleep accompanied by shorter duration of night time sleep, frequent nocturnal awakenings with episodes of night laughter or screaming [6,168,169,170,171,172], pointing to the absence of typical developmental maturation or even a regression in the regulation of the circadian sleep–wake cycle.

EEG with concurrent monitoring of EMG activity (polysomnography) is commonly used to differentiate the sleep stages. In 1968, Rechtschaffen and Kales described the five different sleep stages, with four stages related to slow progression from drowsiness to the deep sleep, characterized by increases in slow wave activity in the EEG (SWS for slow wave sleep) and the paradoxical sleep stage with rapid eye-movements and EEG pattern resembling that in wakefulness (REM sleep) [173]. The percentage of REM sleep within the sleep cycle is abnormally low in patients with RTT [162,174]. In contrast, the percentage of time spent in SWS is enhanced in patients with RTT [174], especially in younger age groups (2–5 years) [162]. However, the absolute duration of overnight SWS is shortened in patients with RTT [141] due to less total time spent asleep. Incidence of seizures in RTT was associated with a significantly lower absolute number of SWS cycles, pointing to more severe problems with SWS efficiency in RTT patients with epilepsy.

The existing literature on sleep EEG in RTT patients is mostly limited by qualitative descriptions of prevalent abnormalities. Three principal EEG changes, although not universally observed in all RTT patients, have been reported during sleep. First, there is a loss of the hallmarks of sleep stage II—sleep spindles and vertex transients that are absent both in young children and adults with RTT [39,79,86]. Second and less frequently observed is the so-called “trace alternant” sleep pattern, whose occurrence during typical development is limited to the neonatal period [175]. This pattern containing high-amplitude bursts of irregular delta activity followed by a relative suppression of EEG amplitudes was described in the early EEG studies in about two thirds of individuals with RTT irrespectively to their age [81,87,162]. Third, epileptiform discharges occur frequently during sleep in the majority of RTT patients [38,68,79,84,88,136,162]. In particular, Aldrich and colleagues compared the spike count at different sleep stages within night sleep and found that occurrence of spike is higher in SWS and in early morning hours [79]. The higher prevalence of spikes in the early morning hours was not due to general increase in SWS in these hours and might be related to circadian variation in cortical excitability and/or hormonal factors.

The only quantitative study of spontaneous EEG activity in RTT patients (*n* = 10) is that of Ammanuel and colleagues [141]. They found a significant increase in delta power during SWS sleep and shortened overall time spent in SWS in RTT girls aged 2–9 years as compared to age-matched control girls. These girls also did not demonstrate a typical decrease of delta power over consecutive SWS cycles during the night. Interestingly, this pattern characterizes the sleep architecture of typical individuals after prolonged sleep restriction [176], and is considered a protective mechanism that compensates for previous sleep loss. Considering the chronic reduction of SWS duration in RTT girls, the same compensatory mechanisms may contribute, among other factors, to their highly increased delta power during SWS and a lack of its reduction at later stages of the sleep cycle. 

To sum up this section, we should point to the need for more qualitative EEG studies of EEG changes during sleep in patients with RTT as they might provide new and more reliable biomarkers of RTT than those registered during active state. Normal sleep is crucial factor for the development of cognitive functions [177,178,179] and this state should not be ignored by researchers of RTT pathophysiology.

#### 2.3.2. Animal Studies

In line with EEG results obtained in patients with RTT, the overall time spent in SWS was significantly shortened in animal models with MECP2 dysfunction as compared to wild type animals [117]. RTT mice were also characterized by a lower number of SWS cycles in combination with a prolonged duration of each cycle [112]. However, changes in sleep delta power in MECP2-deficient mice who reached puberty go in the opposite direction to that observed in 2–9 year old girls with RTT [117]. No conclusions can be drawn from this between-species comparison, due to the small number of relevant studies and major differences in biological age between patients and animals across the studies.

The disturbed sleep–wake cycle found in both RTT patients and animal models can be understood via MECP2′s influence on the circadian clock system. A recent study in mice revealed that MECP2 binds and transcriptionally activates the circadian clock genes, *Per1* and *Per2*, known to play major roles in circadian rhythms [180]. The suprachiasmatic nucleus (SCN) is a master circadian pacemaker controlling the timing of the sleep–wake cycle and coordinating circadian changes in the activity across brain and body tissues. The SCN is mostly active during the light part of the diurnal cycle and mainly shuts down at night. The resetting of SCN cells by light exposure is mediated via intensification of vasoactive intestinal peptide secretion, which is crucial for normal light-induced synchronization of the circadian system [181,182]. In comparison to wild type animals, *Mecp2*-mutant mice exhibit a reduction of vasoactive intestinal peptide expressing neurons of SCN and significant attenuation of the SCN spontaneous activity in the daytime [183]. The *Mecp2*-mutant mice demonstrated increased mortality to a chronic experimentally induced weekly phase shift of the light/dark cycle, possibly as a consequence of disrupted SCN functioning and resulted in sleep deprivation.

In light of animal studies, the discoordination of the circadian sleep–wake cycle that is frequently observed in RTT patients may be caused by disturbed SCN functioning and have severe consequences not only for cognitive functions but also for health status. The decreased spontaneous SCN activity might be also linked to the increased tonic inhibition suggested from other animal studies of MECP2 dysfunction reviewed in previous Section 2.1 (Figure 1). The data also suggest that avoiding any changes in daily routine is of crucial importance for the well-being of patients with RTT. 

### 2.4. Behavioral and EEG Abnormalities in Relation to RTT Genotype

#### 2.4.1. Patient Studies

Recent studies suggested that some *MECP2* mutations might be associated with slightly different manifestations of RTT clinical symptoms [184,185,186,187,188] nonsense mutations in the beginning of the gene (such as R168X, R255X and R270X) and large deletions within *MECP2* associated with more severe RTT phenotype [189,190,191,192]. The relationship between *MECP2* mutation and seizure phenotype has been investigated in several studies with the largest studies of the last decade summarized in Figure 3 [43,44,51,53,192,193]. Noteworthy, unlike general large difference in the reported prevalence of seizures (e.g., close to 90% in the study of Pintaudi [44] and colleagues and less than 40% in the study of Bao and colleagues [53], the profile of changes between genotypes seems to be rather consistent. For example, seizures were more frequently observed in patients with T158M and less frequently in those with R306C mutations across these studies. As Cuddapah and colleagues did not report the percentage of individuals affected with seizures in each genotype group, but characterized the between genotype difference by average scores [192], we did not present their results in the figure. Nonetheless, these authors’ findings are compatible with previous reports as they found that seizures were less frequent in R306C cases as compared to R255X and R294X in a sample of 815 patients with typical RTT [192]. At the same type, there are discrepancies between studies. Patients with the R106W mutation were associated with higher risk for seizures in all but one study ([193]; notably, this was the largest *n* = 1135), and patients with R255X mutations in all but the study of Glaze and colleagues [51]. The reason for this inconsistency needs to be further identified. Probably, these genotypes are particularly sensitive to the methods of seizure assessment. Noteworthy, more clinically severe genotype (e.g., R270X) did not always correspond to phenotypes with higher prevalence of epilepsy, supporting the idea about the non-causative role of epilepsy in RTT development.

Some links between sleep disturbances and *MECP2* mutations were also suggested. Data from patient fibroblasts [183] pointed to the dependence of circadian disturbances on the mutation types in the *MECP2* gene: the phase delay of Per2 rhythms were not found in the fibroblasts from patients with the T158M mutation, but in those with R106W, who also have high prevalence of night sleep disturbances according to the epidemiological study [170]. Considering these findings, it is interesting to compare sleep EEG patterns between patients with these mutations to shed the light on potential neurophysiological distinction between these mutations.

*MECP2* can also be duplicated in human cells, leading to the *MECP2* duplication syndrome. The clinical phenotype of those patients resembles that of patients with RTT and includes infantile hypotonia, intellectual disability, autistic features, progressive neurological declines, choreiform movements and recurrent respiratory infections [194,195,196]. The similarities of clinical symptoms in the conditions with opposite changes in *MECP2* expression highlight the importance of the optimal level of MECP2 for normal development. *MECP2* duplication syndrome has the lowest severity scores, and the greatest median age of seizure onset (6 years) among developmental encephalopathies. The available reports suggest epilepsy in patients with *MECP2* duplication to be mostly drug-resistant [42,58,197]. A recent report suggested that valproic acid (VPA) monotherapy can be efficient in managing epilepsy in these patients [198]. Contrary to patients with classic RTT caused by *MECP2* mutations, in patients with *MECP2* duplication the emergence of neurologic regression coincides with the onset of epilepsy, suggesting that the epileptic process underlying seizures also serves as a trigger for progression of the disease. However, while severity of *MECP2* mutations is linked to the RTT clinical phenotype, *MECP2* duplication size and gene content predict neither presence of epilepsy, seizure type, age of seizure onset or responsiveness to treatment. Mechanisms by which *MECP2* gene duplications modulate the spectrum and severity of neurocognitive phenotype remain unclear [199].

Patients with mutations in *CDKL5* and *FOXG1* genes shares many of the RTT symptoms including relatively normal initial motor-cognitive development, hypotonia, poor or no spoken language, gait abnormalities, intellectual impairment, microcephaly and stereotypic hand movements and historically has been considered an RTT-like disorder. However, there are clear symptomatic distinctions between patients with typical RTT caused by *MECP2* mutations and patients with deficient *CDKL5* and *FOXG1* genes. First, the latter variants are associated with a significantly earlier age of seizure onset. Second, this is a difference in seizure semiology: patients with *CDKL5* mutations had epileptic spasms (45.3%) more often than those with other genotypes (0.4% for typical RTT with *MECP2* mutations, 4.7% for *FOXG1*) [42]. In addition, patients with *CDKL5* mutations are more often characterized by global developmental delay, cortical visual impairment and severely impaired gross motor function, while patients with *FOXG1* mutations are more frequently reported to have postnatal microcephaly, typically associated with corpus callosum abnormalities and marked dyskinetic movements [200,201,202,203]. Thus, it is recommended to classify these disorders as distinct from RTT to secure proper treatment [42].

Clinical assessment of background EEG in patients with *CDKL5* and *FOXG1* mutations and with *MECP2* duplication shows some slowing of brain oscillations, resembling that in patients with *MECP2*-mutations. However, no quantitative analysis was performed to support these observations [58]. Buoni and colleagues studied three girls with mutations in the *CDKL5* gene that caused RTT-like disorder. The authors reported unique background EEG with diffuse high voltage sharp waves of 6–7 Hz, and absence of the typical rhythmic frontal–central theta activity present in Rett syndrome caused by *MECP2* mutations [69]. Keogh and colleagues quantitatively investigated differences in EEG spectra between RTT patients with *MECP2* and *CDKL5* mutations [140]. While no difference in the power spectra characteristics were found, measures of interelectrode coherence differed between groups: *CDKL5* variants showed lower interhemispheric coherence between occipital (O1-O2) and temporal (T3-T4) electrodes, probably related with more frequent left- or right-lateralized epileptiform abnormalities in patients with this rare mutation. However, considering the low number of patients with *CDKL5* mutations (*n* = 4), these findings must be considered with caution.

It is evident that investigation of the relationship of EEG abnormalities with genetic variants of RTT syndrome is an important direction for future research that might provide better understanding of RTT phenotype and lead to more personalized treatment. A large-scale multicenter consortium would greatly facilitate this endeavor.

#### 2.4.2. Animal Studies

The relationship between mutation type and prevalence of a particular type of EEG pattern or epileptiform activity/seizure were not specifically studied in *Mecp2*-deficient animals. However, some influence of the extend of *Mecp2* dysfunction and seizure type might be suggested: absence seizures were typically reported in *Mecp2* knockout mice, while myoclonic jerks were found in animal model of RTT with truncated *Mecp2* in C-terminal. At the same time, all studies, which described handling-induced convulsive seizures, were performed on mice with full or conditional *Mecp2* knockout.

Early onset of seizures is one of the important symptoms of syndromes caused by mutations in the *CDKL5* and *FOXG1* genes, distinguishing them from the Rett syndrome. However, seizure activity is not fully reproduced in animals with mutations in these genes. Strikingly, the pronounced seizures is also absent in the mouse model of the CDKL5 disorder [204], which is characterized by early onset of seizures in almost 100% of all patients with *CDKL5* abnormalities [42,58,205]. *Cdkl5*–/y mice also generally demonstrate normal spontaneous EEG [204]. Nevertheless, in the studies using proconvulsive agents such as kainic acid [206] and NMDA [207], it is easier to induce behavioral seizures and epileptiform activity on the EEG in *Cdkl5* knockout mice compared to wild type animals. Recordings in freely moving *FoxG1*+/− mice showed higher prevalence of high-amplitude spikes on the EEG. Long-lasting spikes were accompanied by immobility but not by behavioral seizures [208]. However, more pronounced behavioral seizures were registered in *FoxG1*+/− mice after kainic acid intake compared to wild-type animals. As described above, in contrast to syndromes that are caused by mutations in genes that play a primary role in the formation of neuronal excitability (for example, Dravet syndrome and *SCN1A* gene), abnormalities in spontaneous EEG and behavioral seizures are observed much less often in *Mecp2*-, *Cdkl5-* and *FoxG1*-deficient mice (reviewed in [205]). All three genes, which are causative for Rett-like conditions, are general regulators of activity of other genes and are not directly involved in neuronal excitability. 

Optimal level of MECP2, CDKL5 and FOXG1 is necessary for the normal functioning of neurons, and its deviation both towards decrease or increase can lead to similar symptoms. There are animal models of *MECP2* duplication syndrome, such as *Mecp2* overexpression in mice [209], and monkey [210] that showed a Rett-like phenotype. *Mecp2* overexpression in these models led to electrographic discharges accompanied by behavioral seizures in mice [209,211]. *Foxg1* overexpression in mice deep-layer neocortical projection neurons also led to higher neuronal excitability assessed by frequency of electrographic high-amplitude spikes and severity of behavioral seizures induced by kainic acid [212]. 

In sum, EEG abnormalities are largely unexplored in animal models of RTT other than *Mecp2*-knockout models, making it rather speculative to talk about some EEG phenotype specific to particular genotype. Further studies are needed to explore this issue. In relation to the very evident distinction between EEG phenotype of *MECP2* abnormalities and caused by *CDKL5*, *FOXG1* mutations, e.g., in an age of seizure onset, their animal models are much more similar and even do not have clear seizure phenotype. Thus, it might be suggested that optimal levels of MECP2, CDKL5 and FOXG1 might influence the E/I balance via some indirect influence that might be modulated differently in humans and rodents.

### 2.5. EEG as a Biomarker of Treatment Efficacy

The ultimate goal of an EEG biomarker is to serve as a monitoring, treatment response or even surrogate endpoint biomarker as described in context of use (COU) FDA statement [213]. EEG characteristics, targeted at particular neurophysiological processes, may provide a tool for assessing effects of clinical trials without relying on overt behavior and lead to individualized tuning of treatment strategy. Below we review currently available studies in patients with RTT that used EEG measures in the evaluation of the effect of pharmacological and behavioral treatments in RTT patients (summarized in Table 4).

***Antiepileptic therapy***: drugs frequently used to control seizure in patients with RTT are valproic acid (VPA, broad spectrum drug with the ability to increase level of GABA by decreasing its degradation), carbamazepine (CBZ, sodium channel blocker, which binds preferentially to voltage-gated sodium channels in their inactive conformation, which prevents repetitive and sustained firing of an action potential), lamotrigine (LTG, blocks voltage-gated Na^+^ channels), levetiracetam (LEV, that binds to SV2A synaptic vesicle protein, which leads to the reduction in the rate of vesicle release) and topiramate (TPM, blocks voltage-dependent sodium and calcium channels and augments activity at GABAa receptors) either as monotherapy or in various combinations [52]. Another emerging option is natural constituents of the cannabis plant (CBD, reduces neuronal hyperexcitability through a unique multimodal mechanism of action), currently in the phase 3 clinical trial to test its efficacy for patients with RTT Epidiolex (NCT03848832). A ketogenic diet (high-fat, adequate-protein, low-carbohydrate diet, possibly acts through increased conversion of glutamate into GABA and activation of adenosine A1 receptors, which affects potassium channels and leads to membrane hyperpolarization), although rarely used in patients with RTT, seems to have a good effect according to rare available studies [223,224,225]. Thus, epilepsy is controlled in RTT mainly through blocking the action potentials by affecting sodium channels and through increase of GABA concentration. Enhancing GABAa mediated inhibition may be a promising mechanism of reducing hyperexcitability caused by *MECP2* disruption.

Pintaudi and colleagues nicely summed previous retrospective studies of different antiepileptic drugs’ effectiveness in RTT and conducted their own retrospective analysis in a cohort of 165 patients with RTT [52]. As a result, they suggested LMG as a drug of first choice especially for patients with later than typical epilepsy onset in RTT (4–5 years of age), followed by VPA and CBZ being the other most effective options. Noteworthy, LMG was shown to increase gene expression of GABAa receptors subunit in primary cultured rat hippocampal cells [226] supporting the role of GABAa decrease in seizure generation in RTT. In the cohort of 110 patients with *MECP2* mutations Huppke and colleagues found CBZ superiority over VPA in treating epilepsy [227]. As both *MECP2* mutations and VPA were reported to induce histones hyperacetylation, the authors suggested that VPA may even in some cases exacerbate one of the underlying mechanisms for seizures in patients with RTT.

Only a few studies have examined the effectiveness of drugs on seizure reduction using prospective studies with EEG recording. Specchio and colleagues performed a prospective pragmatic open-label study to examine the effect of levetiracetam (LEV) in eight girls with RTT, who had drug-resistant epilepsy [214]. LEV significantly reduced seizure frequency in these patients and improved background EEG by decreasing the frequency of epileptiform discharges. Hagebeuk and colleagues performed a prospective randomized, double blinded placebo-controlled crossover study examining the effect of folinic acid in 12 patients with RTT [57]. The results were not very promising: reduction of seizure rate was found only in two out of eight patients who had frequent seizures prior to treatment, whereas three patients developed seizure during treatment; frequency of epileptiform discharges decreased only in three patients but increased in two patients. We suggest that prospective examination of the effects of anti-epileptic drugs on EEG epileptiform pattern and baseline activity in patients with RTT might provide important clinical and scientific insights as the dynamic of neurophysiological changes through antiepileptic therapy is of crucial importance.

***Recombinant human insulin-like growth factor (rhIGF1, Mecasermin)*** is the most studied treatment in RTT patients that has used EEG as an outcome measure [215,216,217]. RhlGFI is suggested to potentiate BDNF, a key target of MECP2 transcriptional regulation, and is considered a potential treatment of RTT disorder. Promising results come from the hIPSC model in which treatment with IGF1 increased the level of KCC2, which as we noted in Section 2.1.2. may play a crucial role in epileptic phenotype in RTT, in MECP2 deficient neurons and restored the timing of the developmental switch of GABA from excitation to inhibition [134]. However, no consistent effect of rhIGF1 on spontaneous EEG characteristics was found. Pini and colleagues [216] reported some decrease in delta-band activity accompanied by increased theta-band activity, while O’Leary and colleagues [215] showed increased delta power with a decrease in beta and gamma relative power after rhIGF1. Concurrently, both studies found that rhIGF1 treatment was followed by slight improvement in stereotypic behavior and social communication, but also by adverse effects, with an increase in hyperventilation index and depressed mood scores [215,216]. At this point meaningful interpretation of the impact of rhlGF1 on normalization of EEG is hardly possible.

Another EEG measure that holds an initial promise to index the effect of rhIGF1 in patients with RTT was EEG frontal asymmetry in the alpha range (8–13 Hz) [217]. Studies on depression and anxiety have found lower right compared to left alpha power over frontal scalp sites, which has been attributed to a predominance of the right hemisphere activation in these conditions [228,229,230]. While mood disorders are not among the primary symptoms of RTT, patients with RTT do show anxiety-like behavior [231]. Three studies examined EEG frontal asymmetry in girls with RTT and monitored the effect rhIGF1 on this measure and on behavioral symptoms [215,216,217]. One study reported right-sided frontal alpha asymmetry in girls with RTT before the open-label extension period of the clinical trial [217]. This abnormal right-hemispheric bias was partially or completely reversed after treatment in five out of the six subjects with RTT. However, neither of these findings were replicated in independent studies [215,216]. The inconsistency in the results on frontal alpha asymmetry in RTT may be due to high interindividual variations of this EEG index in patient samples, questioning its potential clinical application (for a recent meta-analysis see [232]). Thus, this EEG measure may be of little value for assessing the neurophysiological systems subserving mood and anxiety in RTT.

***Glatiramer acetate (GA)*** is a collection of synthetic polypeptides that stimulate secretion of several neurotrophic factors, including BDNF, in the brain. GA was shown to normalize BDNF-level in a rodent model of RTT [233]. Djukic and colleagues performed a prospective open-label trial to assess the effect of GA on gait velocity, cognition, respiratory function, electroencephalographic measures and quality of life in 10 patients with RTT [219]. The authors found improvement in gait velocity, memory novelty score and breathing after GA in girls with RTT. GA treatment also led to a decrease in abundance of epileptiform discharges in the 1-h video EEG recording EEG in all patients who had them at the baseline (*n* = 4), although without a measurable reduction in seizure frequency. Qualitative assessment of EEG by an experienced clinician did not reveal any other treatment-related changes in the EEG patterns.

***Cerebrolysin (CBL)*** is a brain-derived peptidergic drug that includes BDNF and other neurotrophic factors. Gorbachevskaja and colleagues studied the effect of CBL in nine patients with RTT in an open-label trial and reported the amelioration of behavioral symptoms and neurophysiological indexes [137]. Twenty-days of treatment with CBL led to a significant decrease in theta and tentative increase in beta spectral power in the resting frontocentral EEG spectra of patients with RTT, thus reducing differences from the EEG spectra of typically developing peers. The positive effect of CBL on RTT symptoms has also been found in *Mecp2*-deficient animals [234].

***Dextromethorphanpolistirex (DM)*** is a potent noncompetitive antagonist of the NMDA receptor channel. Considering that NMDA-receptors are implicated in pathophysiology of RTT [235,236], Smith-Hicks and colleagues performed a prospective open-label trial of DM and examined its effect on spike activity, seizures and clinical severity in patients with RTT [218]. No EEG parameters other than spike counts were assessed. They found significant dose-dependent treatment-related improvements in clinical seizures, receptive language, and behavioral hyperactivity, while there was no discernible effect on spike counts. More specific EEG characteristics related to NMDA receptor functioning, e.g., the auditory steady-state response [237], are more likely to be sensitive to the effects of DM and can be used in future studies. 

***Transcranial direct current stimulation (tDCS)*** has shown promise in improving cognitive abilities and motor skills of RTT patients. Fabio and colleagues applied anodal tDCS stimulation (C3 electrode site) together with linguistic training in three females with RTT who were previously unresponsive to linguistic training alone [222]. Using a single case design, they showed that tDCS with language training led to significant albeit small improvements in speech sound production (an increase of the number of phonemes produced) and enhanced attentional focus on the task. TDCS combined with a rehabilitation technique also increased the absolute power of theta, alpha and beta bands in patients’ resting EEG. The increase in theta power reflects some worsening of EEG and was not reproduced in a recent tDCS study by the same research team [221] that included 31 RTT patients, and was carefully designed to compare tDCS effects in non-sham and sham control groups. Other results were reproduced and pointed to the improvement of behavioral and neurophysiological functions. Future independent studies are needed to replicate the effect of tDCS empowered by cognitive training on speech production in patients with RTT, and scrutinize potential adverse effects of tDCS, especially in RTT patients with epileptiform activity.

Based on animal studies, it can be assumed that deep brain stimulation might be another efficient way to improve RTT symptoms [238,239]. Behavioral assessment five weeks after brain deep stimulation of the fimbria-fornix showed positive effects on learning and memory, which were accompanied by improved long-term potentiation [238]. After a course of deep brain stimulation adult female *Mecp2*+/− mice became indistinguishable from WT animals when tested in contextual fear conditioning and spatial memory paradigms. The improvements involved only functions with crucial hippocampal involvement, showing specificity for functions related to the stimulated region. The same protocol of deep brain stimulation decreased synchrony between hippocampal CA1 pyramidal neurons to WT-like levels and improved excitability of hippocampal interneurons [239].

***Visual discrimination training***: Fabio and colleagues found that after 5-days training with eye-tracking technique directed at improvement of visual discrimination and categorization abilities, girls with RTT (*n* = 21) unlike control group of RTT girls who were not trained for 5-days (*n* = 13) demonstrated better performance of the same visual tasks that were used in the training sessions [220]. The improvements were manifested in a faster gaze shift to a target visual stimulus and longer duration of fixation on the chosen image. The extent to which such behavioral improvement can be generalized to other images, if the tasks are reasonably similar, remains unclear. Long-term training was accompanied by increase in relative beta and decrease in theta power in few electrode sites. The authors considered these EEG changes as an evidence of normalization of background EEG characteristics related to general brain functioning.

To sum up, there is a lack of well-designed studies to make a reliable general conclusion of treatment efficiency for improving verbal and cognitive skills in patients with RTT. Most of the effects demonstrated are weak or, in some cases, do not even reach statistical significance when corrected for multiple comparisons. While some EEG characteristics were used to assess the effect of treatment in patients with RTT, the results are inconsistent and lack clinical relevance. We believe that future studies should focus more on the EEG characteristics that showed promise in targeting neurophysiological processes affected in RTT. For example, 1/f slope of EEG spectrum or slowing of mu-rhythm. Event-related potential characteristics are also among promising biomarkers for RTT (see review [40]).

## 3. Method

For this review, article selection was conducted according to the preferred reporting items for systematic reviews and meta-analyses (PRISMA) [240]. A flowchart of the selection process is displayed in Figure 4. Firstly, PubMed database were searched using terms ((Mecp2 OR Rett syndrome) AND EEG). All searches were limited to English language, full text publications (excluding conference communications) and performed/updated in April, 2020. This led to 195 unique articles. An additional 7 studies were added that did not come up on the search but were identified in reading the identified literature. Abstract-based screening identified 105 studies reporting EEG characteristics/epilepsy in RTT, including 21 reviews: one review was related to systematic review of event-related potentials in Rett syndrome [40] that is out of scope of the current review; and most others (*n* = 11) focused on epileptiform discharges and seizures in patients with RTT [41,46,54,58,223,241,242,243,244,245,246]. One review was devoted to findings in RTT animal models [205]. One review was focused on non-relevant pathology [247]. The remaining review papers (*n* = 8) only briefly covered EEG abnormalities in patients with RTT, mostly summarizing available clinical evidence [136,248,249,250,251,252,253]. Among them, the most detailed study is that of Glaze and colleagues [136]. After full-text reading, 84 articles were retained for inclusion in our review: 7 articles measuring quantitative EEG characteristics in patients with RTT, 11 articles reporting treatment effects on RTT’s EEG measures, 38 qualitative clinical descriptions of EEG patterns, 15 studies on seizures without reference to the EEG recordings and 11 studies on animal models of RTT. While epilepsy-related studies dominate our search, we put the major focus of our review into quantitative EEG studies trying to bridge results obtained in patients with RTT and in animal models. The following data were extracted from all quantitative and clinical EEG studies in patients with RTT: sample size, age, experimental setting and EEG results (Table 1 and Table 3). Animal studies are summarized in Table 2. For the studies reporting treatment effects on EEG in patients with RTT we added the treatment schema and treatment effect on EEG and on other RTT characteristics (Table 4). 

## 4. Concluding Remarks

Here we reviewed the available data on spontaneous EEG abnormalities in patients with Rett syndrome. We tried to link the identified potential EEG biomarkers of *MECP2*-dysfunction with underlying particular neurophysiological processes via findings from animal studies. Through our review we examined the hypothesis about the dominant role of altered inhibition in the development of RTT. Clinically, E/I balance is usually taxing by seizure and epileptiform activity, while some quantitative measures of spontaneous EEG also show promise, e.g., 1/f spectral slope. At first glance these measures point to the opposite change in E/I balance in RTT: predominance of epilepsy suggests increased E/I balance while steeper 1/f spectral slope points to a decreased cortical excitability. However, animal studies showed that both reduced and enhanced inhibitory transmission may contribute to seizures and/or EEG epileptiform activity in RTT. In particular, there is the region-specific interplay between tonic and phasic inhibition in the development of RTT: heightened extrasynaptic GABAb receptor activation leading to increased tonic inhibition in thalamocortical system and reduced synaptic GABAa receptor activation leading to decreased phasic inhibition mainly in hippocampal circuits.

Available evidence is still inconclusive about the link of EEG abnormalities with the severity of RTT. While antiepileptic drugs efficiently control seizure in about 70% of RTT patients, they do not improve cognitive-motor development, at least at the group level. Thus, epilepsy is hardly an underlying cause, rather it is a common feature of RTT disorder. Unlike clinical seizures, multifocal epileptiform abnormalities in interictal EEG appear to correlate with more severe disturbance of cortical functions in RTT patients, whereas focal centro-temporal EEG spikes may be linked to deficient motor control in speech and movement domains. Prospective quantitative examination of antiepileptic drugs on EEG epileptiform patterns and baseline activity in patients with RTT might provide important clinical and scientific insights and contribute to personalized epilepsy treatment. Predominance of low frequency activity and a relative decrease in the power of alpha and beta rhythms in the background EEG, consistently reported in patients with RTT, was also related to RTT severity. Although the above-described EEG abnormalities are not specific to RTT and characterize several other neurological conditions and other developmental disorders, spontaneous EEG may help to target the specific neurophysiological processes thought to be involved in RTT pathogenesis.

For example, changes in alpha and theta frequencies are known to be valuable biomarkers of the prodromal state of neurodegenerative diseases, such as Alzheimer (e.g., [254]) and Huntington [255] diseases, and correlate with severity of early structural damage [254]. Furthermore, computational modeling suggests that the slowing of alpha rhythms and a simultaneous decrease of alpha band power in the brain may reflect the reduced number of active synapses in the thalamo-cortical cell populations [256]. At the same time, spontaneous occurrence of rhythmic theta in RTT patients, and especially slowing of its peak frequency, according to the *Mecp2*-deficient animal models, may be more specifically linked to the deficiency of catecholaminergic modulation of cortico-hippocampal circuitry in RTT [111]. Another useful quantitative alpha index, which is accessible in RTT patients, is alpha reactivity (open vs. closed eyes). Marked decrease in alpha reactivity is shown in brain disorders, which share some symptoms with RTT and characterized by deficits within the cholinergic system (e.g., Lewy body dementia) [257]. There is strong evidence that cholinergic deficit may be critical for the etiology of social disturbances in RTT animal models [258]. Thus, future studies should include this measure in their analysis.

A shift of the alpha and theta peaks to slower frequencies represent another promising RTT biomarker, which is while frequently reported, was, surprisingly, not studied systematically and quantitatively in RTT. One of the questions that needs to be answered is whether this shift is specific to motor areas and function or represents general phenomena.

Among the EEG parameters that are abnormal in animal models of RTT are high-frequency gamma rhythms, another index of E/I balance. The lack of studies of gamma-band activity in patients with RTT reflects the low sensitivity of scalp recorded EEG to high frequencies mostly due to its heavy contamination by muscle artefacts. Sleep studies might partially overcome this problem. There is also a hope for the MEG technique, which was shown to be able to reliably assess the gamma band activity in spontaneous mode, and in response to auditory and visual stimulation [259].

Several EEG parameters, such as gamma band power in sleep and frontal low-frequency activity when watching movies, showed atypical developmental trajectories in RTT. These findings are very promising as they might indicate the neurophysiological changes underlying disease progression. More studies are needed to examine the validity of these measures to differentiate the disease stages and to be implicated in clinical practice. There are also only few studies on the relationship between EEG abnormalities and genetic variants of the RTT syndrome, and this represents an important direction of investigation, as different mutations associated with RTT can result in different neurophysiological changes and might benefit from a mutation-tuned treatment approach. Large scale multicenter consortiums are needed to further proceed towards establishing EEG biomarkers in relation to age/disease progression and genetic mutations in patients with RTT.

While most knowledge about mechanisms underlying RTT pathophysiology currently comes from animal studies, RTT manifestation even in a validated animal model of RTT is clearly different in some aspects from that of humans. For example, RTT seizure phenotype is not fully reproduced in MECP2-knockout mice. Another promising approach to dig into the mechanisms underlying RTT phenotype is human pluripotent stem cells, which already brought several insights about alternation of cellular processes in RTT. Great progress in creating 3D cultures allows to study EEG-like activity in brain organoids [260,261]. This approach might be crucial to reveal the early signs of RTT and effects of particular mutations. We believe that translatable electrophysiological indexes will help to bridge works on cellular and whole organism levels.

Overall, consideration of the EEG characteristics opens promising directions of research that can bridge studies in animals and patients and provide informative objective index of particular neurophysiological processes that are crucial for understanding RTT pathophysiology and for assessment of treatment efficacy. The hope is to provide the patients and families with more specific counseling concerning the course of the disease, symptom surveillance, prevention and treatment.

## Figures and Tables

**Figure 1 ijms-22-05308-f001:**
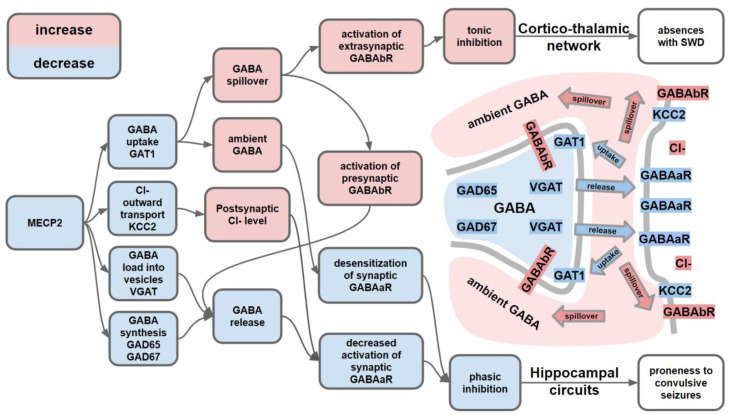
Role of GABAergic impairment in pathogenesis of epileptic activity in the animal model of RTT.

**Figure 2 ijms-22-05308-f002:**
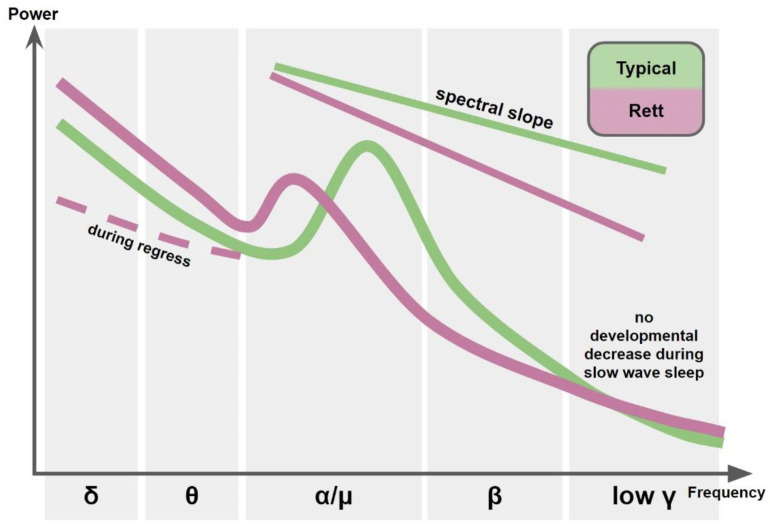
Spectral features of EEG in RTT.

**Figure 3 ijms-22-05308-f003:**
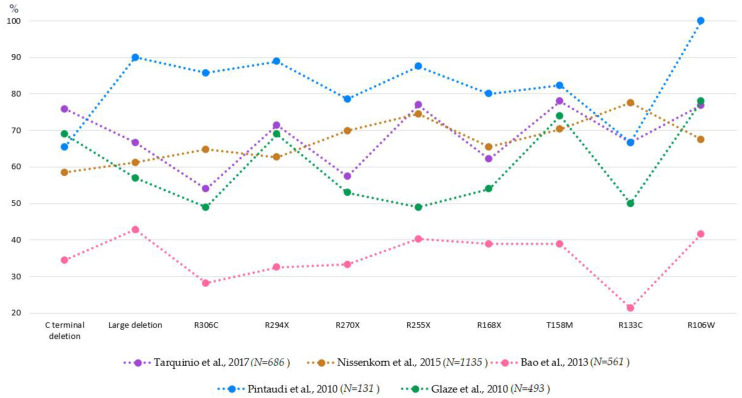
Prevalence (%) of seizures in patients with RTT depending on the particular mutation in the *MECP2* gene.

**Figure 4 ijms-22-05308-f004:**
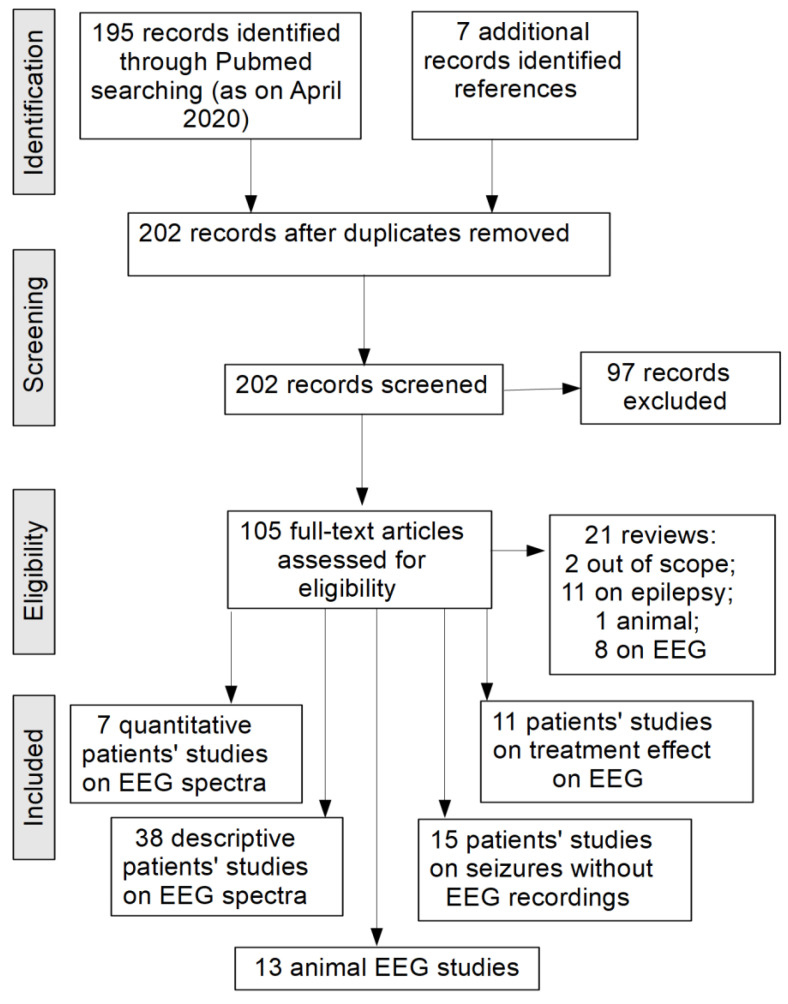
PRISMA flowchart.

**Table 1 ijms-22-05308-t001:** Characteristics of studies with qualitative, clinical description of EEG abnormalities.

Study	Sample (*n*: Age)	EEG Settings	EEG Characteristics
[59]	RTT: 1, 4 y o	Video EEG	Needle-like central spikes evoked by contralateral passive finger-tapping drug-resistant
[60]	RTT: 1, 5 y o	Video EEG (24 h)	Multifocal bilateral discharges precipitated by right-hand tapping lips (but not cheek or abdomen, or the left-hand tapping lips, or observer’s hand tapping the lips) and immediately disappearing when the movement stopped.
[61]	RTT: 1, 12 y o	EEG	Paroxysmal runs of fluctuating 4- to 5-Hz rhythmic frontocentral theta activity at rest that abated with movement or tactile stimulation
[62]	RTT: 1, 4 y o	A 32-channel scalp EEG (24 h)	Centrotemporal spikes (CTS) disappeared after hand clapping
[63]	6 RTT among 74 genetic-dysmorphic syndromes	EEG video monitoring with at least one EEG including awake and spontaneous afternoon nap recording	Abnormal EEG in 3 RTT
[64]	RTT: 1, 7 y o Lhermitte-Duclos syndrome//Neurofibromatosis: 1, 8 y o	Multiple daytime and sleep EEGs	Continuous spike and wave in slow-wave sleep Focal epileptiform discharges while awake
[65]	RTT: 64: 3–9	Awake and sleep video EEG (24 h)	Unilateral, highly rhythmic hand tapping accompanied by contralateral synchronous centrotemporal spikes and not responsive to drugs (*n* = 5)
[56]	RTT: 8: 7–20 y o	Video-polygraphic, EEG + EMG + EKG recording	– Slowing of the background activity – Epileptiform abnormalities (spike and sharp wave discharges) – Abnormal sleep patterns – Epilepsy: drug-resistant in half//misdiagnosed (*n* = 6) – Multifocal and asynchronous cortical myoclonus (*n* = 5), myoclonic seizures (*n* = 4), myoclonic status (*n* = 2)
[66]	RTT: 3: 14, 18, 22 y o	Video-EEG and polygraphy (with confirmed reflex seizures in *n* = 1).	Reflex seizures, triggered by food intake (*n* = 1) or self-provoked by rhythmic pressure on hands (*n* = 2) or by taking someone’s hand or holding on to a table (*n* = 1)
[50]	Zappella RTT *MECP2* high-intermediate performance (HIP): 11: 11–38 y o low-performance (LP): 5: 8–19 y o	8-channel EEG referenced to linked mastoids continuous awake and sleep (at least 30 min)	Centro-temporal spikes: HIP > LP Multifocal EEG activity: HIP < LP EEG encephalopathy: HIP < LP Spindles and K-complex: HIP > LP Epilepsy (recurrent unproved seizures): HIP < LP (ns)
[49]	RTT: 18: 7–21 y o All with seizures	20–30 min video EEG awake	Correlation between epilepsy and behavior: EEG stage III (moderate to marked slowing of background activity with dominant theta and delta activity) EEG stage IV (no occipital dominant rhythm and marked slowing of background activity). (1) theta slow activity over the frontal and central regions (eight cases) (2) frontocentroparietal spikes (eight cases) (3) generalized spike and waves (one case) (4) diffuse subcontinuous spike and waves suggestive of epileptic encephalopathy (one case)
[44]	RTT (165 including Classic (140) Preserved speech variant (PSV) (15) Hanefeld (6) 130 (78%) with epilepsy	Video EEG (not reported) Italian multicenter retrospective study	Epilepsy and RTT variants//mutation type no epilepsy (*n* = 35), drug-responsive (*n* = 81) and drug-resistant (*n* = 49)
[67]	RTT: 2: 7, 12 y o	EEG and lower limb EMG during gait	EMG burst were not associated with clinical jerking but EMG burst-locked averaging of the EEG showed contralateral centroparietal spiking preceding the burst by about 35 ms, indicating a cortical reflex myoclonus.
[45]	RTT (154 including 65% with seizures)	8-channel EEG referenced to linked mastoids continuous awake and sleep (at least 30 min)	Epilepsy and RTT variants//mutation type Drug-resistant epilepsy, DRE (*n* = 16) No relationship between EEG characteristics and DRE
[68]	RTT: 11: 1–33 y o		Seizures (*n* = 8) Epileptiform activity (*n* = 7)
[69]	RTT: 3: 9.5, 7.4, and 9.4 y o, each with a mutation of the CDKL5 gene.	Video EEGs	Seizure onset 1.5 months
[70]	Girl with mutation of MECP2 but no clear RTT phenotype: 8 y o	EEG	5 and 7 y o EEG: presence of high-amplitude delta waves with a notched appearance and a persistent theta activity over posterior regions (EEG of Angelman Syndrome)
[71]	RTT: 50: 1–14 y o	16-channel EEG	Descriptive EEG, epilepsy presence is not reported Changes with RTT progression
[72]	RTT: 1: MECP2 mutation At age 2 and 6	EEG	Age 6: rhythmic triphasic 2- to 3-Hz, high voltage (200–500 mkV) activity, mixed with spikes or sharp waves, with a maximum over the frontal regions (EEG of Angelman Syndrome)
[73]	RTT: 10: < 5 y o Angelman Syndrome (AS):10 Mental retardation:10		Central and/or centro-temporal spike-wave complexes as specific to RTT
[74]	RTT: 191 (detailed survey) 78–EEG	Awake and sleep EEG	76% clinical seizures 78% epileptiform activity that preceded seizure onset
[75]	RTT: 13: 2–17 y o	Awake and sleep EEG, SPECT	Epileptiform activity (*n* = 10) Frontal hypoperfusion that is not correlated with EEG abnormalities
[76]	RTT: 10: 2–16 y o The reported 10 cases were selected because of their peculiar slow EEG rhythms	The EEG studies were performed in wakefulness (whenever possible) and sleep, mostly induced with moderate dosages of chloral hydrate. Passive movement carried out whenever spikes or central theta activity occurred.	CTS/abnormal theta reduced after hand movement (*n* = 4)
[77]	RTT: 16: 8 months-20 y o 44 EEG	EEG, respiration	Pseudoperiodic pattern, the short bursts of high-amplitude slow waves tending to be associated with apnea and the lower-amplitude faster rhythms with normal breathing or with hyperventilation (*n* = 8)
[78]	RTT: 14: 6–17, mean 7 y o TD: 12: 6–18, mean 14 y o	Day time video records, respiration	General EEG abnormality with excess of polymorphic slowing and poorly developed (daytime) sleep change (all *n* = 14) Unreactive Theta (*n* = 10) Absence of normal slow-wave response to hyperventilation (all *n* = 14) Attacks of vacancy and staring not associated with significant EEG changes (*n* = 6) No-epileptic slow-waves mostly during normal breathing (*n* = 11)
[79]	RTT: 4: 4–11 y o	All-night electroencephalograms (EEGs)/polysomnograms on 2 consecutive nights	Epileptiform activity maximum over 1–2 SWS sleep stage and in the morning hours
[80]	RTT: 4: 3.5, 6, 11 and 12 y o	Light no-REM sleep or the state lethargy (wake without slight index of awareness)	Epileptiform activity, in particular, CTS, blocked or attenuated by passive finger movements
[81]	RTT: 30: 2–22 y o 127 EEG-recordings	Awake and sleep EEG, EMG	Epileptiform activity (*n* = 26) CTS (*n* = 10) Pseudoperiodic delta bursts (*n* = 13) Flat record (*n* = 2) Developmental changes
[82]	RTT: 13: 2–17 y o	Awake and sleep EEG, EMG	Seizures (*n* = 5) Epileptiform activity (*n* = 10) Pseudorhythmic flattering (*n* = 4)
[83]	RTT: 8: 2–16 y o	Awake and sleep EEG	Background EEG slowing Epileptiform activity A monotonous theta rhythm (MTR), which was not influenced by either opening or closing of the eyes but attenuated only by a big noise or strong pain stimuli, characteristically dominated the waking tracing EEG through disease progression
[84]	RTT: 52: 1–13 y o 83 EEG recordings	Awake and sleep 8-channels EEG, EMG	Seizures (*n* = 26 + 3) Epileptiform activity, enhanced during sleep (*n* = 43) CTS often coincided with stereotyped hand wringing or tapping and facilitated over the contralateral central regions by passive tactile stimulation (9 of 26 girls tested) Excesses of theta and delta activity (*n* = 12, *n* = 7)
[39]	RTT: 9: 1–6 y o 22 EEG recordings	EEG, video EEG in two patients	Seizures (*n* = 7) Epileptiform activity (*n* = 9, in 3 epileptiform activity preceded development of seizures) Abnormalities increases with age
[85]	RTT: 7: 1–7 y o	Awake and sleep EEG	Reactive Theta, Excessed Delta, Flattering EEG, epileptiform activity, atypical sleep EEG CTS (*n* = 4) including those evoked by tactile stimulation (*n* = 2) Changes with RTT progression
[86]	RTT: 18: 1–17 y o	Awake and sleep EEG, respiration biogenic amine metabolites	Reduced % stage REM (*n* = 15) EEG slowing Epileptiform activity Absent vertex transients and spindles during NREM
[87]	RTT: 11: 4–14 y o	Awake and sleep EEG	Slowing of background EEG (*n* = 11) activity while awake Multifocal spike-waves (centrotemporal regions) (*n* = 9) Intermittent, high-amplitude discharges followed by relative attenuation of background activity during sleep (*n* = 6)
[88]	RTT: 9: 2–15 y o 35 EEG recordings	EEG	Seizures (*n* = 7) Epileptiform activity (*n* = 7) Slowing of background EEG Progression through disease
[89]	RTT: 8: 4–13 y o	Awake and sleep EEG, EMG	Myoclonic jerks (*n* = 8) EEG is poorly organized with high amplitude slow waves short, non-periodic bursts; focal or diffuse spike and wave complexes increased in sleep
[90]	RTT: 1, 2 y o	EEG	Unspecific modification on EEG, no seizures

**Table 2 ijms-22-05308-t002:** Characteristics of EEG and epileptic phenotype in animal models of RTT.

Study	Genotype (Age)	Electrode Position, Behavioral State	EEG Characteristics	Seizure Characteristics (Prevalence)
[113]	*Mecp−/+* (from 13 to 104 weeks)	hippocampal CA1 and contralateral somatosensory cortex, free behavior		cortical discharges, behavioral freezing 13 weeks average number per hour: 20.7 ± 11.7 duration: 0.9 ± 0.28 s 104 weeks average number per hour: 116.8 ± 44.2 duration: 1.7 ± 0.23 s aggravated by levetiracetam reduced by valproic acid and acetazolamide suppressed by ethosuximide
[116]	*Vglut2-Mecp2flox/y* (10 weeks) *Vglut2-Mecp2LSL/y* (25–30 weeks)	right frontal cortex and somatosensory cortex left hippocampal CA1 and dentate regions		*Vglut2-Mecp2flox/y* (37.5%) cortical spike-and-wave discharges average number per hour: 3.3 ± 2.1 duration: 4.1 ± 0.4 s *Vglut2-Mecp2LSL/y* (25%) cortical spike-and-wave discharges average number per hour: 3.3 duration: 4.5 ± 0.8 s
[110]	*PV-Mecp2-/y* *SOM-Mecp2-/y*			*SOM-Mecp2-/y* (50%) epileptic seizures starting at 12 weeks generalized tonic-clonic seizures observed during routine handling PV-Mecp2-/y no seizures
[111]	*Rosa26-Esr/Cre- Mecp2 Stop/+*, (around 36 week) *Rosa26-Esr/Cre-Mecp2Stop/y*, (5.5–8.5 weeks)	hippocampal CA1 and contralateral somatosensory cortex, free behavior	▼ gamma (35–60 Hz) power (*Rosa26-Esr/Cre-Mecp2Stop/+*; *Rosa26-Esr/Cre-Mecp2Stop/y*) rescue in *Rosa26-Esr/Cre-Mecp2Stop/y* ▼ theta frequency rescue in *Rosa26-Esr/Cre-Mecp2Stop/y*	*Rosa26-Esr/Cre-Mecp2Stop/y* cortical discharges average number per hour: 45 ± 8 duration: 2.8 ± 0.7 s frequency: 6.2 ± 0.3 Hz *Rosa26-Esr/Cre-Mecp2Stop/y-rescue* cortical discharges average number per hour: 21 ± 3 duration: 1.0 ± 0.1 s frequency: 6.8 ± 0.2 Hz *Rosa26-Esr/Cre- Mecp2 Stop/+* cortical discharges average number per hour: 76 ± 17 duration: 1.11 ± 0.1 s frequency: 7.9 ± 0.4 Hz *Rosa26-Esr/Cre- Mecp2 Stop/ + -rescue* cortical discharges average number per hour: 32 ± 4 duration: 1.2 ± 0.1 s frequency: 7.7 ± 0.3 Hz
[117]	*Mecp2tm1.1Bird* (6–7 weeks)	the M1 region of the frontal cortex, free behavior	▼ delta power during NREM	cortical discharges, behavioral freezing 13 weeks average number per 20.7 ± 11.7 versus 116.8 ± 44.2 104 weeks
[98]	*Emx1-Mecp2*, *Dlx6a-Mecp2* (6–8 weeks)	frontal and parietal cortex		*Emx1-Mecp2* spike-wave discharges, absence seizures average number per hour: 36 ± 7 duration: 1.3 ± 0.1 s *Dlx6a-Mecp2* no discharges
[109]	*Mecp22lox/y; Dlx5/6-Cre* *Mecp22lox/y; Emx1-Cre Mecp2Stop/y; Emx1-Cre* (8, 7 weeks) *Mecp22lox/y; PV-Cre* (13 weeks) *Mecp22lox/y; SOM-Cre* (13 weeks)	hippocampus		*Mecp22lox/y; Dlx5/6-Cre* behavioral seizures following handling spikes and wave discharges, behavioral arrest *Mecp22lox/y; Emx1-Cre* no behavioral or electrographic seizures *Mecp2Stop/y; Emx1-Cre* behavioral seizures *Mecp22lox/y; PV-Cre* no behavioral seizures *Mecp22lox/y; SOM-Cre* no behavioral seizures
[118]	*Mecp2−/+*, *Mecp2−/y*, *TH-Mecp Stop/+* (around 36 weeks) *TH-Mecp2Stop/y* (5.5–8.5 weeks)	hippocampal CA1 and contralateral somatosensory cortex, free behavior	▼ theta frequency partial preservation in TH-Mecp2Stop/y ▼ gamma (35–60 Hz) power partial preservation in TH-Mecp2Stop/y	*Mecp2−/y* (100%) cortical discharges average number per hour: 42.1 ± 7.9 duration: 2.8 ± 0.7 s frequency: 6.2 ± 0.3 Hz *TH-Mecp2Stop/y* (100%) cortical discharges average number per hour: 20.5 ± 6.3 duration: 2.2 ± 0.6 s frequency: 6.5 ± 0.36 Hz Mecp2-/+ cortical discharges average number per hour: 69.6 ± 15.5 *TH-Mecp Stop/+* cortical discharges average number per hour: 67.1 ± 13.9
[29]	*Mecp2T158A/y* (4.2 weeks and 12.9 weeks), *Mecp2−/y* (12.9 weeks)	hippocampal electrode, free behavior	▲ gamma high (70–140 Hz) power seizure behaviors (unspecified)	
[112]	*Mecp2−/+* (42–57 weeks)	parietal cortex, free behavior	▼ the average number of delta cycles over a 24-h period ▼correlation coefficient for delta power and movement	cortical discharges, behavioral freezing average number per hour: 10.7 ± 1.6 duration: 0.76 ± 0.01 s frequency: 8.6 ± 0.02 Hz
[119]	*Viaat-Mecp2−/y* (9 weeks) *Dlx5/6-Mecp2−/y* (39 weeks)	cortical electrodes, free behavior		*Viaat-Mecp2−/y* hyperexcitability discharges, no electrographic seizures *Dlx5/6-Mecp2−/y* no hyperexcitability discharges
[107]	*Mecp2−/y* (7–10 weeks), *Mecp −/+* (35–52 weeks)	hippocampal CA1 and contralateral somatosensory cortex, free behavior	▼ theta frequency (hippocampus) desynchronized, low-amplitude cortical activity =cortical delta and hippocampal sharp-wave activity during sleep	*Mecp −/+* (100%) cortical discharges, immobile animal, no myoclonus average number per hour: 9.1 ± 1.3 duration: 1–2 s frequency: 8.8 ± 1.3 Hz suppressed by ethosuximide
[106]	*Mecp2308/y* (age of recording not specified)	frontal and parietal cortex, free behavior		spike-and-wave discharges, behavioral arrest, generalized myoclonic jerks

*Mecp2308/y*-expression of truncated protein, *PV-Mecp2-/y*-deletion of *Mecp2* from parvalbumin positive neurons), *SOM-Mecp2-/y*-deletion of *Mecp2* from somatostatin positive neurons), *Mecp22lox/y; Dlx5/6-Cre*-deletion of *Mecp2* from GABAergic neurons in the forebrain, *Mecp22lox/y; Emx1-Cre*-deletion of *Mecp2* from glutamatergic neurons in the forebrain, *Mecp2Stop/y; Emx1-Cre*-preservation in glutamatergic and glia, *Mecp22lox/y; PV-Cre*, *Mecp22lox/y; SOM-Cre*, *Vglut2-Mecp2flox/y*-*Mecp2* deletion from excitatory neurons, *Vglut2-Mecp2LSL/y*-restoration of expression of *Mecp2* only in excitatory neurons, *Viaat-Mecp2−/y*-deletion of *Mecp2* from >90% GABAergic neurons), *Dlx5/6-Mecp2−/y*-deletion of *Mecp2* from GABAergic neurons in the forebrain, *Emx1-Mecp2*-deletion of *Mecp2* from excitatory neurons and glia in forebrain and hippocampus, *Dlx6a-Mecp2*-deletion of *Mecp2* from inhibitory neurons in the forebrain), *TH-Mecp Stop/+*, *TH-Mecp2Stop/y*-preservation of *Mecp2* expression in catecholaminergic neurons, *Mecp2tm1.1Bird*-*Mecp2* knockout.

**Table 3 ijms-22-05308-t003:** Quantitative EEG studies in patients with RTT.

Study	Sample (*n*: Age)	EEG Settings	EEG Characteristics
[139]	RTT (57 including 20 in active regression (AR), 29 post-regression (PR)): 2–11) TD (37: 2–11)	128-channel Hydrocel Geodesic Sensor Net System, 5–10 min while watching movies	Spectral power (2–50 Hz) ▼ beta (13–30 Hz) at frontal and central sites (absolute and relative) ▼ beta (13–30 Hz) at temporal sites (relative) ▼ high alpha (9–13Hz) widespread (absolute and relative) ▼ low alpha (6–9 Hz) at temporal and occipital sites (relative) ▼ low alpha (6–9 Hz) at central sites (absolute and relative) ▲ delta (2–4 Hz) widespread (relative) ▼ 1/f slope (more negative, in general and in PR) for AR: ▼ theta (4–6 Hz) at frontal sites (absolute) ▼ high alpha and beta (9–30 Hz) at frontal and central (absolute and relative) ▼ high alpha (9–13 Hz) at occipital (absolute) Abnormal development of delta and theta
[140]	RTT, MECP2 (32) PSV, MECP2 (4) RTT, CDKL5 (4) including Resistant Epilepsy (8), Treatment Responsive Epilepsy (16), without Epilepsy (18) Mean age: 7.69 +/− 5.22	8-electrode EEG, minimum of 20 min recordings during resting state	=spectral power (1–50 Hz) ▼ network activity (principal component of inter-electrode coherence) in CDKL5 compared to MECP2 ▲ left-hemispheric predominance in the occipital region Resistant > Epilepsy>no Epilepsy within RTT * ▲ left-sided parieto-occipital coherence in PSV compared to classic RTT *
[141]	RTT (10: 2–9) TD (15: 2–9)	Overnight (F3, C3, and O1), slow wave sleep (SWS)	Spectral power (0.5–45 Hz), ▲ delta Absence of developmental decrease of gamma (35–45 Hz) * ▼ %SWS
[138]	RTT (38(20 with disease onset <1 year and 18 > 1 year): 2–8) TD (60: 2–8)	16 channels; >15.4-s epochs	Spectral power (1.5–25 Hz) ▼ alpha ▲ theta activity in RTT ▼ central and occipital alpha ▲central and frontal theta activity in the group 1 (early disease onset).
[137]	RTT (9:2–8, stage 3) TD (17: 2–8)	16 channels: eyes open (*n*= 6) and eyes close (*n*= 5); >8 4-s epochs	Relative Power (1–20 Hz) ▼ alpha and beta ▲ theta band
[142]	RTT (1:10) TD (10:10)	16 channels: 20 artifact-free 5-s epochs	Spectral power (1.5–30 Hz) ▲ delta and left anterior theta ▼ posterior alpha 1
[143]	RTT (9: 10–22) Compared to unspecified normative data	*n* = 11: 16–20 channels; 5–10 min of recording	Spectral power (1.0–16 Hz): ▲ low frequency activity (unspecified, statistical values not provided)

**Table 4 ijms-22-05308-t004:** Treatment effect on EEG in patients with RTT.

Study	Sample (n: Age)	Treatment Schema	EEG Settings	EEG Effects	Other Treatment Effects
[57]	RTT 75% *MECP2* (12: 2–11, 20, 30)	Folinic acid: 2-year prospective randomized, double blinded, placebo-controlled crossover	Video 16-channel EEG (10–20 system)	Improvement: ▼ Seizure rate (only in 2 out of 8 patients who had seizures at baseline) ▼ Frequency of epileptiform discharges (only in 3 patients) Worsening: 3 patients developed seizure during treatment ▲ Frequency of epileptiform discharges (2 patients)	N/A
[214]	RTT 100% MECP2 with drug-resistant epilepsy (8:8–18)	levetiracetam (LEV) prospective, pragmatic, OLE: mean dose achieved was 44.84 mg/kg/day SD 18.02	EEG at baseline and 6-months after beginning of treatment	Improvement: ▼ Seizure rate, background EEG, epileptiform abnormalities ▼ Frequency of epileptiform discharges (in 6 out of 8 patients)	N/A
[215]	RTT 100% *MECP2* post-regression (17:2–10)	rhIGF-1: 20 wk, double-blind cross over: 0.12 mg/kg/twice a day	128 channels; 5–10 min calm recording when girls watched movie of their choice	= Frontal alpha asymmetry (FA) Worsening: ▲ Frontal delta power ▼ Beta relative power ▼Gamma relative power	Worsening: ▲ The Kerr severity scale, ▲ ADAMS Depressed Mood subscale, ▲ Visual Analog Scale ▲Hyperventilation Improvement: ▼ ABC-C stereotypy ▲ CSBS-DP Social
[216]	RTT 100% *MECP2* stage III (10: 3–11) RTT untreated (9: 2–12)	rhIGF-1: 20–24 wk: 0.1 mg/kg/twice a day	10 channels; 1–2 h EEG	= Frontal alpha asymmetry (FA) Improvement: ▲ Theta frequency ▼ Delta amplitude	Improvement: ▼ International Scoring System: ISS), ▲ RSS: Rett Severity Score ▲ Endurance to RSS testing
[217]	RTT 100% *MECP2* (6 out of 10:2–10)	rhIGF1: 4 wk MAD + 20 wk OLE: 0.120 mg/kg)	Frontal and parietal channels	Frontal alpha asymmetry (FA) (Pre-OLE: R > L with trend to reverse Post-OLE)	Worsening: ▼ Behavioral subtotal (MBA) Improvement: ▼ Apnea index
[218]	RTT 100% *MECP2* with epileptiform activity 33 (13 + 12 + 8 dose groups): 2–14.5)	Dextromethorph an: 25 wk OLE 3 dose groups: 0.25 mg/kg/day; 2.5 mg/kg/day; 5.0 mg/kg/day;	Neuroscan EEG system; Overnight recordings	=Epileptiform spikes and sharp waves during non-REM sleep Improvement: Seizure rate (parental report): low dose group (in all? small number of patients who had seizures before OLE *n* = 5/4/2 at each dose group)	Improvement: ▲ Receptive Language (5.0 mg/kg/day vs 0.25 mg/kg/day) ▲ * Behavioral hyperactivity (2.5 mg/kg dose)
[219]	RTT 100% *MECP2* (10:10–21) ambulatory	Glatiramer acetate (GA): 24 wk of 20 mg/day	A 21-channel 1-h video EEG (wakefulness (*n* = 10), drowsiness (*n* = 5), sleep (*n* = 3))	=Background EEG abnormalities =Seizure rate Improvement: ▼ Frequency of epileptiform discharges (in all 4 patients who had them at baseline)	Improvement: ▲ Gait velocity ▲ Memory Novelty Score % ▼ Breath holding index/%
[137]	RTT N/S (9:2–8, stage 3) 17 TD aged-matched	Cerebrolyzin (CL): 20 days (2.0–2.5 mL daily, i.m.).	16 channels: eyes open (*n*= 6) and eyes close (*n*= 5); >8.4-s epochs	Improvement: ▲beta activity in the parietal region and occipital alpha in the 8–9 Hz narrow band ▼ central and frontal theta the after CL	Improvement: Qualitative assessment by clinicians and parents ▲ the behavioral activity ▲ attention level ▲ motor functions ▲ non-verbal social communication
[220]	RTT (33: 21 + 13: 5–36)	Visual discrimination training: Short-term, 30 min (STT: *n* = 13 + 21) Long-term, 5 days, (LTT: *n* = 21 vs. Control: *n* = 13)	19 dry electrodes (wireless): visual discrimination task, at rest eyes open	After STT: ▼ Alpha and Beta Brain Symmetry Index (shift from L > R into R > L) After LTT: =Brain Symmetry Index Improvement Theta relative power (frontal left and parietal) ▲ Beta relative power (frontal right and parietal left)	After STT: No changes Improvement after LTT: ▲ Length of fixation (FL) ▼ Time before first fixation (TFF)
[221]	RTT (31: 13–35) non-sham tDCS (*n* = 18) vs. sham tDCS (*n* = 13)	TDCS stimulation on Broca’s area together with linguistic training for 20 min/day over 10 days	21 EEG channels	▲ Frequency and Power of Alpha and Beta =Frequency and Power of Theta	Improvement: ▲ Attention ▲ Language abilities (the number of phonemes produced)
[222]	RTT *MECP2* mutations: C468G, R133C, R306C (3: 29, 30, 31) with chronic language impairments	TDCS stimulation on Broca’s area together with linguistic training, case-study design for 20 min/day over 10 days	19-channels EEG (20 min wit eyes closed)	▲ Frequency and Power of Alpha, Beta and Theta	Improvement: ▲ language abilities (the number of vowel/consonant sounds and words and the production and comprehension through discrimination) ▲ motor coordination (functional movements)

## Data Availability

Data sharing not applicable.

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
