# Peer review of "Reviewing Evidence for the Relationship of EEG Abnormalities and RTT Phenotype Paralleled by Insights from Animal Studies"

_ijms, 2021, doi:10.3390/ijms22105308_

Round 1
Reviewer 1 Report
In this paper, the authors reviewed the available data present in the literature on spontaneous EEG abnormalities in patients with Rett Syndrome. They tried to link the identified potential EEG biomarkers of MECP2 dysfunction with underlying particular neurophysiological processes via findings from animal studies. Through their review, they examined the hypothesis about the dominant role of altered inhibition in the development of RTT. They concluded that the EEG characteristics open promising directions of research that can bridge studies in animals and patients and provide an informative objective index of particular neurophysiological processes that are crucial for understanding RTT pathophysiology as well as for assessment of treatment efficacy.
The manuscript contains interesting remarks, is well written and organized and with a very updated bibliography. The quality of the tables is very good and exhaustive.
The conclusions are presented in an appropriate manner and are strongly supported by the data.
There is just one point that must be considered before the publication. The reading of the review is rendered long by the repetitive style used. I think that it would gain considerably if it would be organized in a more efficient way.
This is especially true for example for the part describing the phenotype of the mouse models for Rett syndrome.
Finally, the article is written in standard and clear English.
Author Response
We kindly thank you and reviewers for the comments on our ms. entitled “Reviewing evidence for the relationship of EEG abnormalities and Rett Syndrome phenotype paralleled by insights from animal studies” by Smirnov KS, Stroganova TA, Molholm S, Sysoeva OV. We implemented the suggestion into the revision we are submitting together with this letter.
In response to reviewer 1 we have restructured the ms. and deleted some repetitions.
- Deleted the section 4. Seizures and EEG abnormalities through developmental changes as it contains some repetitions with other parts of the text. Some parts of this section were places in other parts of the ms. (e.g. Introduction, 2.1.1.3. Development of seizures and severity of RTT symptoms; 2.1.1.4. EEG abnormalities in relation to developmental stage etc);
- Deleted some parts in Introduction that we elaborate more in the main text of the ms. (e.g. lines 63-73);
- Shorten the introduction into section 5.Behavioral and EEG abnormalities in relation to RTT genotype;
- We also added new summary figure on EEG spectral characteristics in patients with EEG (Figure 2) to get the reader better representation of the findings.
Reviewer 2 Report
The authors performed a very detailed review on spontaneous EEG abnormalities in Rett Syndrome patients. They tried to dissect relationship between EEG abnormalities and MECP2 mutation. This is very interesting. I may suggest to describe a neglected aspect of this disease. Neurons and progenitors cells coming from animal model of Rett syndrome and from patients are prone to senescence. This may greatly impair neuronal functionality.
Author Response
We kindly thank you and reviewers for the comments on our ms. entitled “Reviewing evidence for the relationship of EEG abnormalities and Rett Syndrome phenotype paralleled by insights from animal studies” by Smirnov KS, Stroganova TA, Molholm S, Sysoeva OV. We implemented the suggestion into the revision we are submitting together with this letter.
In response to reviewer 2 we added the description of the neglected quickly developing field in the study of mechanisms underlying RTT – study on neurons and progenitors’ cells coming from animal model of Rett syndrome and from patients. We believe this is particularly important direction for the future research that already brought some advances into the filed.
See additions into lines 92-95, 1141-1145, 1370-1379.
Round 2
Reviewer 2 Report
none